# In vitro Cas9-assisted editing of modular polyketide synthase genes to produce desired natural product derivatives

Kei Kudo [1,6], Takuya Hashimoto [1,6], Junko Hashimoto [2], Ikuko Kozone [2], Noritaka Kagaya [1], Reiko Ueoka [1], Takehiro Nishimura [3], Mamoru Komatsu [4], Hikaru Suenaga [1], Haruo Ikeda [4✉] & Kazuo Shin-ya [1,3,5✉]

One major bottleneck in natural product drug development is derivatization, which is pivotal for fine tuning lead compounds. A promising solution is modifying the biosynthetic machineries of middle molecules such as macrolides. Although intense studies have established various methodologies for protein engineering of type I modular polyketide synthase(s) (PKSs), the accurate targeting of desired regions in the PKS gene is still challenging due to the high sequence similarity between its modules. Here, we report an innovative technique that adapts in vitro Cas9 reaction and Gibson assembly to edit a target region of the type I modular PKS gene. Proof-of-concept experiments using rapamycin PKS as a template show that heterologous expression of edited biosynthetic gene clusters produced almost all the desired derivatives. Our results are consistent with the promiscuity of modular PKS and thus, our technique will provide a platform to generate rationally designed natural product derivatives for future drug development.

[1] National Institute of Advanced Industrial Science and Technology (AIST), 2-4-7 Aomi, Koto-ku, Tokyo, Japan. [2] Japan Biological Informatics Consortium (JBIC), 2-4-32 Aomi, Koto-ku, Tokyo, Japan. [3] Technology Research Association for Next Generation Natural Products Chemistry, 2-4-7 Aomi, Koto-ku, Tokyo, Japan. [4] Kitasato Institute for Life Sciences, Kitasato University, 1-15-1 Kitasato, Minami-ku, Sagamihara, Kanagawa, Japan. [5] Biotechnology Research Center, The University of Tokyo, 1-1-1 Yayoi, Bunkyo-ku, Tokyo, Japan. [6] These authors contributed equally: Kei Kudo, Takuya Hashimoto. ✉email: ikeda@ls.kitasato-u.ac.jp; k-shinya@aist.go.jp

Recently, middle-weight molecular compounds have attracted increasing attention as new compounds to fill the gap in molecular weight between macromolecules, such as antibodies, and low-molecular-weight compounds. In particular, as Fabrizio Giordanetto et al. reported[1], macrocycles can attack large surface of target, such as protein–protein interaction (PPI), due to their size and complexity. Both their incomplete rigidity and moderate flexibility influence the overall structure of macrocycles, which are essential properties for drug development, resulting in suitable membrane permeability[2] and desirable overall pharmacokinetics[3,4]. Even today, natural products are of great importance as drug leads[5]; however, the significant decrease in the rate of the discovery of novel compounds has become a big problem. In addition, difficulties in derivatizing macrocyclic natural compounds by chemical synthesis due to their structural complexity are also a bottleneck in drug development[6]. Therefore, various congeners will be produced from known useful natural compounds by genetic manipulation.

Among natural products, macrolides, such as erythromycin and rapamycin, are considered good models of macrocycles. These compounds are biosynthesized via an assembly line process that is catalysed by modular polyketide synthases (PKSs) (Supplementary Fig. 1). Despite the usefulness of macrolides in drug development, the genetic modification of modular PKS gene clusters is extremely difficult. The multi-modular architecture of PKSs may have evolved from a single module PKS by gene duplication events[7–9]. Accordingly, the sequences of the modules in modular PKS clusters are highly homologous (more than 70% on average). With these contexts, many have tried to modify type I PKS to produce new analogues of targeted polyketides and proposed several genetic strategies, which still have potential limitations on applying them to larger type I PKS compounds[10–13]. Among these reports, Aleksandra Wlodek et al. recently published a recombination-based methodology to derivatize rapamycin[10]. However, the desired compounds could not be obtained since the homologous recombination occurred so randomly across the PKS modules that could not be under controlled. Alternatively, Robert McDaniel et al. manipulated 6-deoxyerythronolide B synthase by utilising a pair of unique restriction sites as an editing point[11]. The availability of suitable restriction sites is key to their methodology; however, the usage of restriction sites limits its broad applicability since it becomes increasingly difficult to find or introduce artificial restriction sites as the number of modules increases.

For the in-will production of new polyketide skeletons by applying former or even future knowledge to PKS engineering, the credible editing of all type I PKS genes is necessary. However, since the well-known methods described above have obstacles due to the high sequence similarities among PKS modules/domains, we developed a methodology that employs the CRISPR-Cas9 reaction in vitro to avoid unnecessary off-target reactions and simplify sequence analyses of edited constructs. To precisely engineer against highly repetitive modular PKS genes, the establishment of a precise DNA digestion and subsequent seamless cloning method that does not depend on recombination or restriction digestion is of great importance. Herein, we report the in vitro module editing technique and its application to produce the desired rapamycin derivatives.

## Results

**Concept of in vitro module editing**. Based on the heterologous expression system we established that is applicable to biosynthetic clusters with over 200 kb cloned into the BAC vector[14], we developed the idea of in vitro module editing, which applies a combination of both CRISPR-Cas9[15] and Gibson assembly[16] to

the BAC clone in vitro (Fig. 1a). Once we construct vectors harbouring artificial gene clusters, they can be expressed in suitable hosts, such as *Streptomyces avermitilis* SUKA strains[17], to validate the productivity. Because the heterologous expression experiment does not allow unexpected mutations anywhere in the biosynthetic genes, this methodology guarantees accurate editing, which is its most important advantage. As a proof of concept, we selected rapamycin (**1**) (Supplementary Fig. 1) to establish the in vitro module editing system since some rapamycin derivatives[10] can be index for target derivatives (for a review of chemically synthesised derivatives of rapamycin, see ref. [18]). If we can accurately edit the modular PKS of rapamycin, whose modules show higher homology to each other (the average homology between 14 KS domains is 91.6%.) than those of other systems such as the erythromycin PKS (86.1% homology between 6 KS domains), this technique will be applicable for most natural products.

We established the heterologous production of rapamycin in *S. avermitilis* SUKA strains, since the production yield of rapamycin was high enough that rapamycin could be isolated, and was better in these strains than in other host strains. The biosynthetic gene cluster for rapamycin from *S. rapamycinicus* NRRL5491[19] was cloned into the BAC vector pKU503 to yield pKU503rap (166 kb, accession number: LC566301). In addition, the expression of the positive transcriptional regulator gene, *rapH*[20], in the host strain (designated SUKA33::*rapH* or SUKA34::*rapH*) led to the greatest production of rapamycin (8.6 mg l$^{-1}$) under our conditions (Supplementary Fig. 1).

**Acyltransferase domain-targeted module editing**. The minimum PKS module contains a set of three domains, a ketosynthase (KS), an acyltransferase (AT), and an acyl carrier protein (ACP), that catalyse a 2-carbon chain elongation (Supplementary Fig. 1). The sequence of AT domains controls the selection of methyl (alkyl) malonyl-CoA or malonyl-CoA as the extender units, which determines the α-branched or non-branched substructures. Our first aim was to change the extender unit for a particular module of rapamycin PKS by replacing the AT domain with AT domains from the different modules of the same rapamycin PKS (i.e., the AT domain of module 6 for the methylmalonyl extender unit and the AT domain of module 8 for the malonyl extender unit. See also Supplementary Fig. 2).

We first targeted module 9 to replace a malonyl-CoA unit with a methylmalonyl-CoA unit. The corresponding position is the triene moiety of rapamycin which is reported to play a significant role in the biological activity[21,22]. To precisely cut out the nucleotide sequence encoding the AT domain in module 9 (M9AT) of the BAC clone pKU503rap, the CRISPR-Cas9 system was employed in vitro (Fig. 1b). A pair of sgRNAs was designed to target the region flanking M9AT (Supplementary Tables 1 and 2). The sequence alignment of the PKS modules was used to avoid the homologous region, especially the 3′ regions of target sequences, which may cause off-target cleavage in other modules. As a result, a band for the digested BAC segment and another band corresponding to the targeted 1.6 kbp segment were observed by gel electrophoresis (Fig. 1c). The digested BAC segment was then subjected to Gibson assembly with a donor DNA fragment. The donor DNA fragment was prepared by overlap PCR and consisted of an exchanged domain (i.e., an AT domain from another module) flanked by unchanged sequences (Fig. 1b and Supplementary Tables 1 and 2). The M9AT region with the KS-AT linker and AT-DH linker was replaced with the corresponding region of M6AT (Fig. 1b and Supplementary Fig. 3). The desired clone was screened by colony PCR (Supplementary Tables 1 and 2), and the sequence of the edited region was confirmed by Sanger sequencing (Fig. 1d).

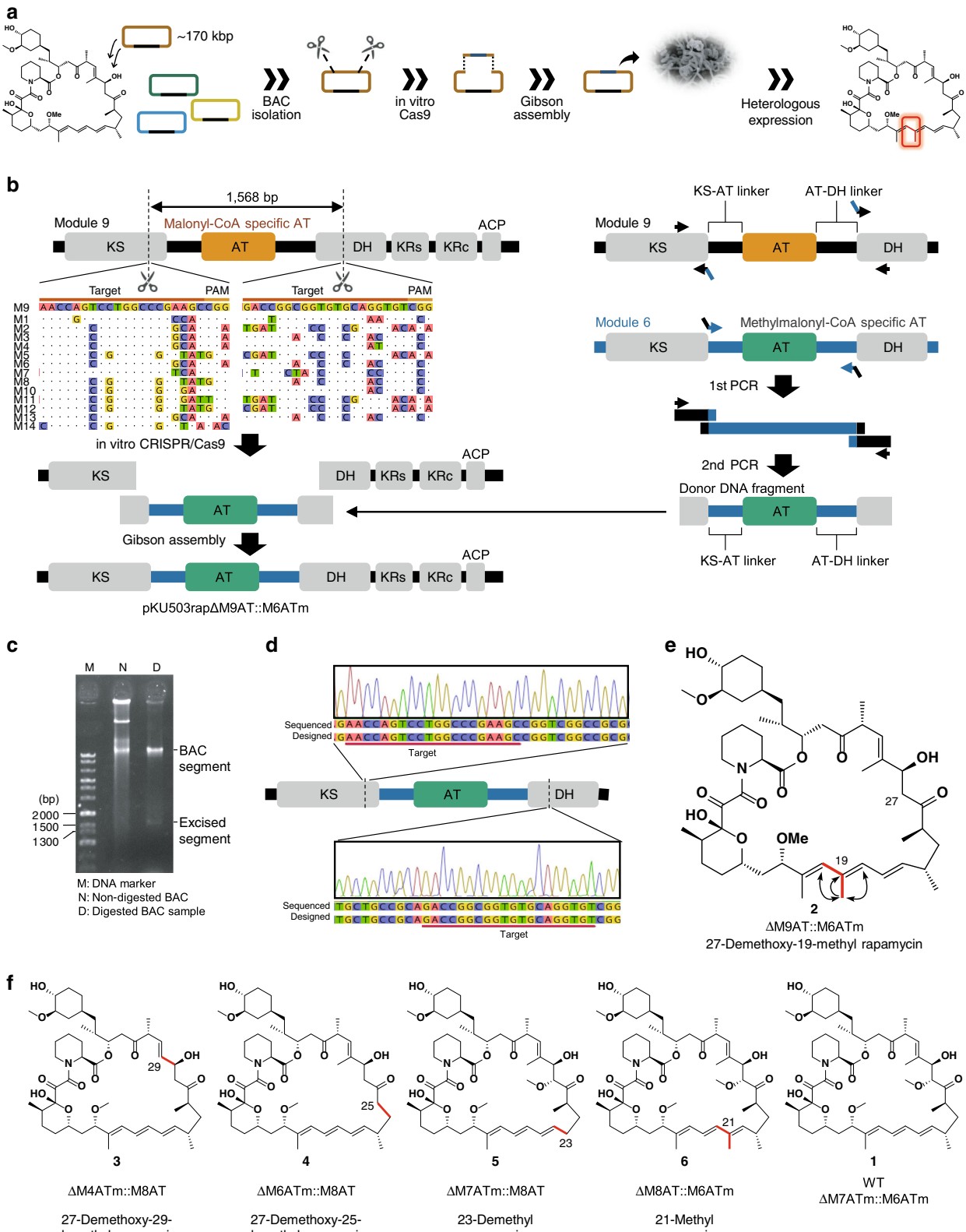

**Fig. 1 The scheme for in vitro module editing and AT-exchange. a** Concept art of in vitro module editing. **b** Practical workflow for the M9AT-exchange experiment. The target sequences of sgRNAs and the equivalent regions of repeated modules within the rapamycin PKS are aligned. The donor DNA fragment was prepared by overlap PCR of three fragments. Dotted lines indicate the excised position. **c** Gel electrophoresis of Cas9-digested pKU503rap. The experiments were performed five times independently with similar results. **d** Sequence confirmation of pKU503rapΔM9AT::M6ATm. **e** Structure of the product for pKU503rapΔM9AT::M6ATm. Arrows indicate the observed HMBC correlations. **f** Other AT-swapped rapamycin derivatives produced in this study.

The BAC vector pKU503 harbouring the edited artificial rapamycin biosynthetic gene cluster (pKU503rapΔM9AT::M6ATm) was first transformed into *S. lividans* TK24 carrying the linear plasmid SAP1, which was then transferred into *S. avermitilis* SUKA34::*rapH* by exogenous conjugation[23]. As a result of cultivation, a UV absorption peak (280 nm) consistent with a triene was observed, as well as an MS peak corresponding to dehydroxy rapamycin (920.5489 ($C_{52}H_{82}NO_{13}Na^+$, $[M + Na]^+$: −0.6 mmu)) (Supplementary Fig. 2) in the HR-ESI-MS data. Finally, the structure of the produced compound was confirmed by analysis of its NMR spectra such as HMBC and DQF-COSY (Supplementary Data 1–7). Analyses of the $^1H$ spin couplings together with the $^1H$-$^{13}C$ long-range couplings, especially key HMBC correlations from a new methyl proton signal to C-18, C-19 and C-20, revealed that the methyl residue was attached to C-19. The oxymethine carbon, C-27, in rapamycin was replaced by a methylene carbon in this rapamycin derivative. These analyses established the structure as 27-demethoxy-19-methyl rapamycin (**2**) (Fig. 1e). This result indicated that the PKS function was altered as intended. In the same manner, we performed a series of AT exchanges targeting modules 4, 6, 7 and 8 (Supplementary Fig. 2). As a result, we succeeded in constructing each desired clone and observed the MS peaks and the MS/MS fragmentation patterns corresponding to 27-demethoxy-29-demethyl rapamycin (**3**), 27-demethoxy-25-demethyl rapamycin (**4**), 23-demethyl rapamycin (**5**) and 21-methyl rapamycin (**6**) (Fig. 1f and Supplementary Figs 2 and 4), all of which are novel rapamycin analogues. Compound **6** was isolated from scaled-up culture, and its chemical structure was confirmed by NMR analyses (Supplementary Data 8–14). These results indicate that the substrate tolerance of the PKS domains allows them to accept both α-methylated and α-demethylated biosynthetic intermediates.

**Production of module deletion and insertion derivatives**. Following the production of AT-exchanged derivatives, we turned to module deletion (Fig. 2a) and insertion (Fig. 2b) to further examine the promiscuity of PKS modules towards upstream ACP-bound substrates. We carried out single module deletion processed targeting modules 6, 8 and 10 (Fig. 2a and Supplementary Figs 3 and 5). Artificial constructs designated pKU503rapΔM6, pKU503rapΔM8 and pKU503rapΔM10 were prepared and introduced into SUKA strains. As a result, MS peaks corresponding to three new 29-membered rapamycin derivatives (**7**, **8** and **9**) were observed (Fig. 2c and Supplementary Fig. 6). The structures of the compounds **7** and **9** were determined by NMR analyses (Supplementary Data 15–28), and the production of compound **8** was much lower than that of **7** and **9** (Supplementary Table 3).

Using our in vitro module editing technique, we also confirmed the production of derivatives in the previous research[10] to prove whether we can rationally target specific regions. Unlike their results of accidental recombination products, we accurately prepared pKU503rapΔM3, pKU503rapΔM2ACP-ΔM4KR, pKU503rapΔM3KS-ΔM6KS and pKU503rapΔM2AT-ΔM8AT, which corresponded to ISOM-4280, 4144, 4185 and 4193 (producing 29-, 27- and 25-membered rings and acyclic rapamycin analogues), respectively. As a result of the heterologous expression in SUKA33::*rapH*, the production of three of the four derivatives (**10**, **11** and **12**, but not the 25-membered ring rapamycin analogue) was confirmed by the combination of HR-ESI-MS and NMR analyses (Supplementary Fig. 6 and Supplementary Data 29–49).

For the expansion of the macrocycle, we inserted module 12 between modules 2 and 3 to biosynthesize a 33-membered ring derivative **13**, which is the same compound as that produced by

ISOM-4309[10]. Since the length of the donor DNA fragment was much greater than the length of one module unit (~11 kbp), the donor DNA fragment was divided into five segments that could be amplified by PCR, and they were concatenated by Gibson assembly to afford the desired sequence. Using this long donor DNA fragment led to the construction of pKU503rap_M2-M12-M3 (Fig. 2b). As shown in this example, in vitro module editing can accurately generate the intended products even with large gene units. As expected, this module insertion derivative was the same as that produced in Aleksandra Wlodek's study (Supplementary Fig. 6 and Supplementary Data 50–56).

**Editing of reductive loop domains**. Depending on the module, there may be additional processing enzymes, namely, a ketoreductase (KR), a dehydratase (DH), and an enoyl reductase (ER). This set of contiguous domains is called a reductive loop[24], and it regulates the oxidation level around the β-keto groups of nascent chains (Supplementary Fig. 1). The manipulation of reductive loops leads to dramatic changes in the structure of the product and often impacts the biological activities of the macrolide compounds. Therefore, we focused on editing the triene moiety. We targeted module 7 to make a tetraene moiety, as follows. We constructed two clones; one was the loss-of-function mutant of the NADPH binding site in the ER domain in which GGVGMA was replaced with SPVGMA[25], and the other was the mutant with the entire ER domain deleted. The KR domain consists of a structural subunit (KRs) and a catalytic subunit (KRc). The ER domain resides in the linker between KRs and KRc, as shown in Fig. 3a[26]. To construct the M7ER-deleted mutant, the M7ER region was replaced with the corresponding linker in M8KR.

The resultant constructs, pKU503rap_M7ER[0] and pKU503rapΔM7ER, were transformed into SUKA54, and the culture extract was monitored by UPLC-TOF-MS. A peak with a UV absorption characteristic of a tetraene moiety was observed, and its MS signal corresponded to the desired new rapamycin derivative (**14**) (Fig. 3c and Supplementary Fig. 7). The structure of **14** was established by NMR spectroscopy (Supplementary Data 57–63).

We next tried to remove the conjugated triene functionality and produce a 1,5-diene moiety by targeting the olefin at the centre of the triene substructure (Fig. 3c). The DH-KR didomain of module 9 was swapped into the DH-ER-KR tridomain of module 7 (Fig. 3b). Based on previous studies, we exchanged the two reductive loops between the portion immediately after the conserved RYW sequence at the AT-DH linker and the (A/V)(Q/R)W sequence at the KR-ACP linker (Fig. 3b)[24]. The resulting construct, pKU503rapΔM9DH-KR::M7DH-ER-KR, was expressed in SUKA34::*rapH*, and the extract of the culture was monitored by UPLC-TOF-MS. The peak showing a UV absorption characteristic of a triene moiety disappeared, while the HR-ESI-MS results were consistent with the reduced rapamycin derivative (**15**) (Supplementary Fig. 7). As expected, NMR analysis revealed the chemical structure of **15** to be 19,20-dihidro-rapamycin (Supplementary Data 64–70), which to our knowledge is a newly isolated rapamycin derivative.

In addition, we replaced the inactive DH-ER-KR tridomain of module 3 with the active one of module 7 to produce 32-desketorapamycin (**16**) (Fig. 3c) which could not be obtained by the homologous recombination-based system[10]. The construct, pKU503rapΔM3DH-ER-KR::M7DH-ER-KR, was prepared, as well as pKU503rapΔM9DH-KR::M7DH-ER-KR, and was expressed in SUKA34::*rapH*. Analysis of the culture extract using UPLC-TOF-MS indicated the product possessed distinctive rapamycin UV absorption and had a consistent *m/z* value of **16** (Supplementary Fig. 7). The chemical structure of compound **16**

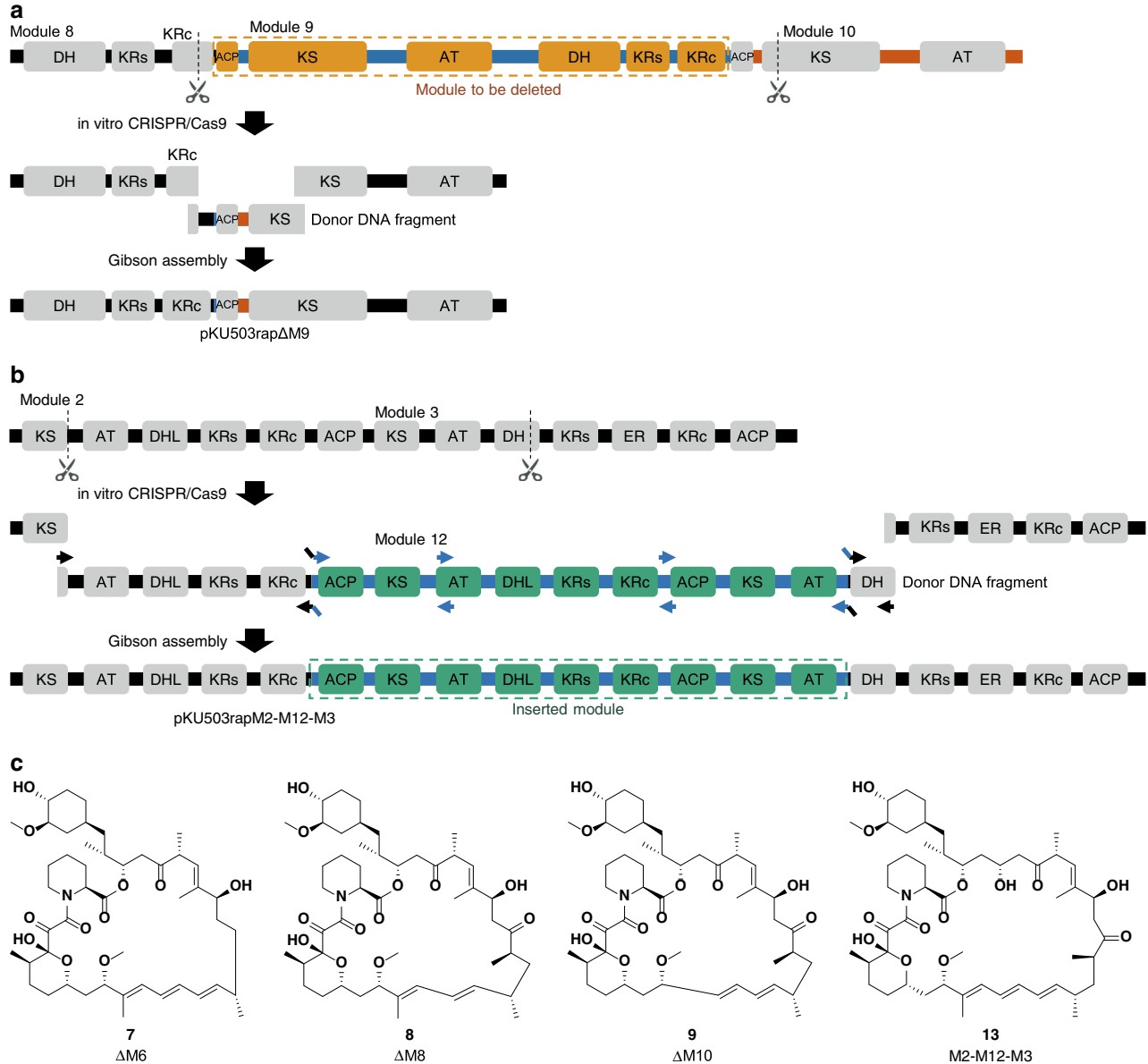

**Fig. 2 Module deletion and insertion.** Schematic representations of module deletion (**a**) and insertion (**b**). **c** Rapamycin derivatives produced in this study.

was further confirmed by NMR analysis (Supplementary Data 71–77). Compound **16** was chemically synthesised from rapamycin via reduction of the ketone at the C-32[27], and have been tried clinical application of coronary stent therapy[28].

**Inversion of stereochemistry and double modification**. Based on the successful exchange of the reductive loop, we tried to change the absolute configuration of a hydroxy functional group. The configuration of the β-hydroxy groups is controlled by the KR domains which are classified into A-type and B-type[29]; an A-type KR domain results in the L configuration, while a B-type KR domain affords the D configuration at the β-position of the ACP-bound nascent precursor. Since all the KR domains in rapamycin PKS are B-type domains, we adopted A-type KR domains from other macrolides to invert the hydroxy groups of rapamycin. We targeted the KR domain in module 11 (M11KR) for replacement by two A-type KR domains, the bafilomycin KR domain from module 2 (bfmM2KR) and the leucomycin KR domain from module 7 (lcmM7KR) (Fig. 4a). The interchange points were the

same as those in the cases of pKU503rapΔM9DH-KR::M7DH-ER-KR (Supplementary Fig. 3).

Based on the results of the heterologous expression in SUKA34::*rapH*, these artificial biosynthetic gene clusters were revealed to produce a rapamycin derivative that showed a triene-derived UV absorption band and an HR-MS and MS/MS signal consistent with demethyl rapamycin (Supplementary Figs 7 and 8). The NMR spectra indicated that the planar structure of this derivative is 16-*O*-demethyl rapamycin (Supplementary Data 78–84). The stereochemistry of **17** was determined as (16*R*)-16-*O*-demethyl rapamycin since **17** was identical to the chemically prepared (16*R*)-epimer (Supplementary Fig. 9, for more detail, please see Determination of the configuration of **17** section in Supplementary Information)[30].

To optimise candidate compounds for drug development, modifying several substructures in the skeleton is often necessary. Therefore, we attempted to simultaneously edit multiple modules. The combined modification construct involving both the ER domain dysfunctional mutation of module 7 (pKU503rap_M7ER[0],

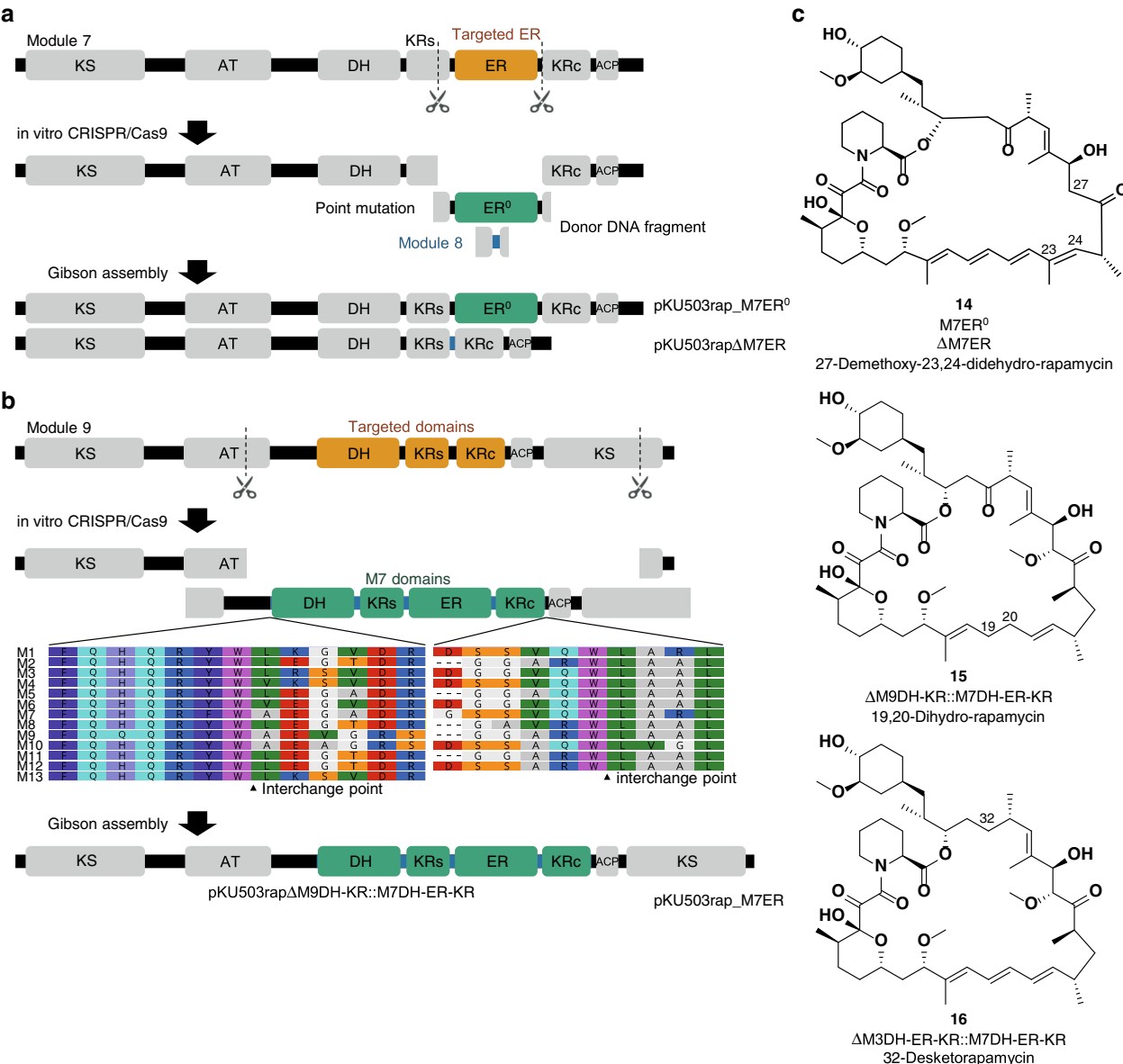

**Fig. 3 Schematic representation of the swapping of the reductive loop domains. a** Constructions of the ER-inactivation and ER-deletion mutants. **b** Construction of the reductive loop-swapped mutant. **c** Structures of the rapamycin derivatives produced in this section.

see the former section) and the stereochemical inversion mutation of M11KR was prepared via replacement with bfmM2KR. As a result of the heterologous expression of this double-modified construct in SUKA34::*rapH*, while its production was relatively low (Supplementary Table 3), two rapamycin derivatives showing tetraene-derived UV absorption bands were detected. Based on the MS/MS analyses (Supplementary Fig. 8), these derivatives were deduced to be 27-demethoxy-23,24-didehydro-rapamycin (**18**) and 27-demethoxy-16-*O*-demethyl-23,24-didehydro-rapamycin (**19**). This result suggested that type-I PKS is highly flexible enough to accept multiple editing. Compound **18** was deduced to have a methoxy group at the C-16 position, while the corresponding methylated derivative of compound **17** was not observed. The responsible methyltransferase, RapM, was reported to accept various alkyl group donors other than *S*-adenosyl methionine[31]. While the relaxed specificity of RapM against polyketide substrates may describe this contradiction, further biochemical and structural analyses are required.

**Biological activities**. Rapamycin shows its immunosuppressive activity via enhancing the interaction between FKBP12 and mTOR resulting in the formation of three components complex (FKBP12-rapamycin-mTOR)[21]. We employed fluorescent based PPI detecting technology Fluoppi (Fluorescent based technology detecting Protein-Protein Interactions) analysis to evaluate the binding activities of rapamycin derivatives simultaneously to FKBP12 and mTOR. Fluoppi harbours a tetrameric fluorescent protein hAG tag and an oligomeric assembly helper Ash tag to create detectable fluorescent foci in cells when there are interactions between two proteins fused to each tag[32]. As the results, **15**, in which the triene substructure was replaced by a 1,5-diene moiety, showed potent binding enhancing activity with comparable level to that of rapamycin (**1**) (Supplementary Fig. 10 and Table 1). This result was interesting since the triene moiety in rapamycin is considered to play a significant role for the binding enhancing activities[21]. This, to our knowledge, is the first report of the potency of 15, although its detailed mode of action

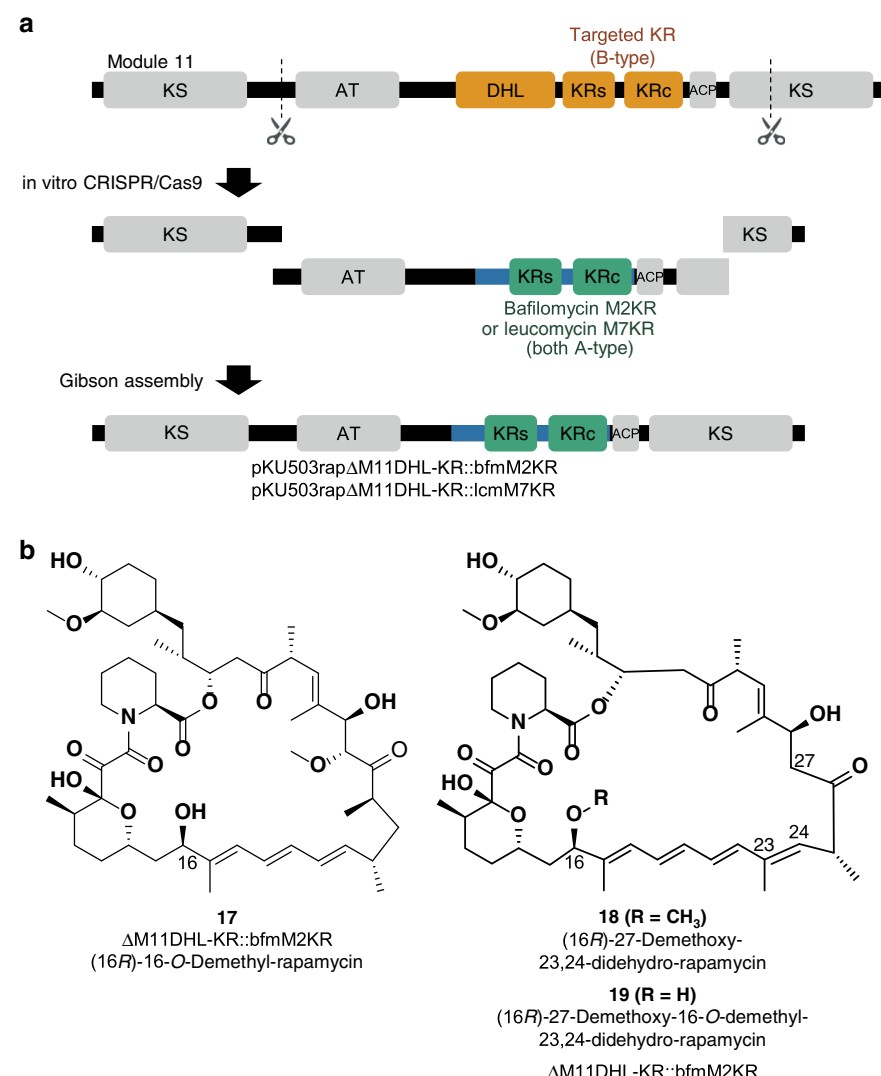

**Fig. 4 Inversion of the stereochemistry of one of the hydroxyl groups. a** Schematic representation of the swapping of the reductive loop of module 11 with the corresponding region of module 2 from bafilomycin PKS. **b** Structures of **17**, **18** and **19**.

| Table 1 Biological activities of rapamycin and its derivatives to induce the formation of FKBP-mTOR complex. | |
|---|---|
| | EC$_{50}$ (nM) |
| **1** | 9.0 |
| **2** | 45 |
| **7** | 320 |
| **9** | 34 |
| **10** | 21 |
| **11** | n/a |
| **13** | 73 |
| **14** | 2130 |
| **15** | 6.3 |
| **16** | 38 |
| **17** | 58 |

EC$_{50}$ values are calculated from three independent biological replicates. n/a, not active. Source data are provided as a Source Data file.

mechanisms are remains unclear. The other derivatives, **2**, **9**, **10**, **13**, **16** and **17**, showed relatively weaker activities for the induction of PPI between FKBP12 and mTOR than that of **1** (Table 1). Notably, **7** and **14** were much less potent than **1**, and **11** had

completely lost the ability to induce the formation of FKBP12-mTOR complex. It seemed that the geometrical changes of the whole macrocyclic ring structures reflected the effect of these derivatives, however, further analysis is required to design a derivative with desired activity.

In addition, we also performed the cytotoxic studies of these compounds against T lymphoma Jurkat cells, human ovarian adenocarcinoma SKOV-3 cells and malignant pleural mesothelioma MESO-1 cells, since rapamycin was reported to exhibit cytostatic effects against cancer cell lines[33]. As shown in Supplementary Fig. 11, cells treated with rapamycin (**1**) and most of its derivatives gradually reduced their viabilities and kept plateau levels at middle range indicating the cytostatic effects, while etoposide, utilised as a positive control, showed typical cytotoxic activities. In the dose-response curve of **1** and **16**, plateau levels for Jurkat, SKOV-3 and MESO-1 cells were around 25%, 60% and 75%, respectively. This is consistent with the previous literature[34] that **16** has comparable activity to **1**. Compounds **17**, **7**, **13**, **9**, **15** and **14**, in ascending order of the minimal inhibitory concentration, slightly suppressed the cell viabilities. Compounds **2**, **10** and **11** showed cytotoxic activities against the tested cell lines only at unusually high concentrations (over 10 μM) implying that biological activities of these

compounds were weakened. In addition, the PPI-inducive activities and the cytostatic effects seemed to have no significant relationship each other (Table 1 and Supplementary Fig. 10). For example, contrary to the potent PPI inductive activity, the cytostatic effects of **15** was ~100-fold lower than that of **1** and was almost the same magnitude of cytostaticity as that of **14** of which PPI-inducive activity is quite low. Thus, the derivatization by engineering of rapamycin PKS has potential to lead other type of active compound, proving the usefulness of module editing techniques.

## Discussion

While the in vivo CRISPR/Cas9 system is commonly employed in genome editing in eukaryotic cells (especially in mammalian cells), it is not applicable for the engineering of bacterial modular PKSs due to the occurrence of unpredictable recombination events. To date, only one application of the in vitro Cas9-assisted gene editing has been reported to manipulate the biosynthetic gene[35]. Yunkun Liu et al. developed a strategy for gene deletion and marker gene insertion on a cosmid vector of which size is up to 40 kb[35]. For another example, the combination of in vitro Cas9-digestion of the genomic DNA and Gibson assembly is capable of capturing up to 100 kb of targeted biosynthetic gene cluster[36]. In this study, we demonstrated that our in vitro module editing could introduce target DNA fragments from endogenous (rapamycin) and/or exogenous (bafilomycin and leucomycin) PKS genes into desired sites precisely even when the BAC exceeds 160 kb. Since modular PKSs are among the most difficult targets, this methodology should be applicable to other large biosynthetic genes, such as most other PKSs and non-ribosomal peptide synthetases. In the conventional method, the derivatives of the target compounds were produced by the wild-type producer. Many strategies for genome editing of *Streptomyces* using CRISPR/Cas9 technology in vivo have been developed[37,38]; however, it is still challenging to introduce a large DNA fragment into non-model host strains and to avoid off-targets especially in the PKS genes. In contrast, our approach is based on the high probability of heterologous expression in a versatile host. One of the most important characteristics of our system, which uses heterologous expression, is that we can produce a variety of derivatives of the target compounds with the same platform.

Engineering modular PKS enzymes has been a long-standing goal. The manipulation of PKS modules may lead to the loss of the intimate domain/module interactions required for precise nascent chain transfer, elongation reactions and β-keto processing[39] (Supplementary Fig. 1). Moreover, in the traditional understanding of the PKS reaction cascade, KSs are considered strict gatekeepers that recognise specific substrates for the chain elongation reactions[39]. However, from our experimental results on rapamycin along with the linear polyketides aureothin[12] and monensin[13], substrate promiscuity is broad enough to accept various unnatural intermediates. Although there are still few successful examples of designed PKS, these cases are undoubtedly setting the stage for utilising the substrate tolerance of modular PKSs. These prospects encouraged us to establish a theoretical combinatorial biosynthesis for the production of desired polyketides based on this design, which includes computer simulations to find ways to increase the biological activities. By employing in vitro module editing, we can assess any designed genetic manipulation of a natural product without significant limitation. Further investigations on the module boundary and non-native interactions between exogenous modules have been conducted to conceptualise the combinatorial biosynthesis.

Recently, drug screenings have shifted from target-based screenings to phenotypic screenings since the candidates obtained by target-based screening have low successful rate to become clinical drugs than those expected. In addition, more first-in-class drugs have been discovered from phenotypic screenings than from target-based screenings[40]. As recently developed advanced techniques, superior phenotypic screening models that have not been used previously, such as patient-derived iPS cells and patient-derived tissue organoids, can be employed. In these phenotypic models, natural products often afford better results. However, the identification of the exact target molecule is essential in modern drug development. Due to the difficulties associated with target identification, it has become a bottleneck in natural product-based drug development. Our technique simplifies the introduction of new functionalities such as hydroxy groups at desired positions for synthesising labelled derivatives that can act as bait. These scaffolds can also be used to synthesise the payload of antibody-drug conjugates. Very recently, 27-O-desmethyl rapamycin derivative was reported to show specific activity against mTORC1, indicating the derivatization of potent natural product such as rapamycin could induce alternative activity[41]. Therefore, this technique is likely to become indispensable in natural product-based drug development. We are currently succeeded in applying this technique to not only other PKS compounds, pikromycin and leucomycin, but also a non-ribosomal peptide compound, YM-254890, indicating that this technique can be widely available for the derivatization of natural compounds. The paradigm that the substrate specificity of PKS is flexible now become common in derivatizing the skeleton of natural products.

## Methods

**Bacterial strains and culture conditions.** The growth of *Escherichia coli* strains DH5α, NEB10β (New England Biolabs Japan Inc., Tokyo, Japan) and GM2929 *hsdS*::Tn10[42] was achieved using LB containing 10 g of tryptone, 5 g of yeast extract and 5 g of NaCl in 1 l of deionized water adjusted to pH 7.5, and the solid medium (LA) was prepared by adding 15 g of agar to 1 l of LB medium. In the production of rapamycin and its derivatives, *Streptomyces avermitilis* SUKA carrying the biosynthetic gene cluster for each derivative was inoculated into a 50-ml test tube containing 15 ml of GSY medium containing 5 g of glucose, 15 g of soya flour, and 5 g of yeast extract in 1 l of deionized water, and the cells were cultured with reciprocal shaking at 320 rpm at 27 °C for 2 days. A 375 μl aliquot of the vegetative culture was used to inoculate a 125-ml flask containing 15 ml of production medium containing 40 g of β-cyclodextrin, 20 g of Pharmamedia, 5 g of glycerol, 21.2 g of MES, 5 mg of $ZnSO_4 \cdot 7H_2O$, 5 mg of $CuSO_4 \cdot 5H_2O$, and 5 mg of $MnCl_2 \cdot 4H_2O$ in 1 l of deionized water adjusted to pH 6.0. The fermentation culture was carried out with a rotary shaker at 24 °C for 5 days at 180 rpm. Large-scale preparations of the products were carried out in a baffled 500-ml flask containing 100 ml of the production medium.

**Genome sequencing and construction of the BAC library.** Genome sequencing of the *S. hygroscopicus* NRRL5491 strain was performed using a PacBio RS II (Pacific Biosciences, Menlo Park, CA), and the sequence data were assembled using HGAP2 (Pacific Biosciences). A BAC library of *S. hygroscopicus* NRRL5491 was constructed following a previously reported protocol. Each of the obtained BAC clones was stored in a 384-well plate containing Plusgrow II (Nacalai Tesque, Kyoto, Japan) (100 μg ml⁻¹ ampicillin and 20% glycerol) at −80 °C. A clone carrying the targeted gene cluster was screened by PCR amplification using two pairs of primers (rapF1: AACAGCCGAAAGAAATGGCTGTGC/rapR1: GGCCCTCT CGAACTTCCGTACCTC and rap F2GGTGGTTTCGTCATGCCTGTTCTG/ rapR2 GCTCTCCTTGAGCATCAGCCACTG) to amplify the upstream and downstream regions of the gene cluster, respectively. A clone, pKU503rap, was selected, and the inserted sequence was confirmed by end-sequencing.

**Introduction of edited BACs into *S. avermitilis* SUKA.** All clones were introduced into *S. lividans* TK24 carrying the SAP1 vector (containing the synthetic sequence of attB$_{\phi K38-1}$, attB$_{R4}$, attB$_{\phi BT1}$, attB$_{\phi C31}$ and attB$_{TG1}$ of *S. avermitilis* MA-4680). All clones were integrated into the SAP1 vector because the bacteriophage attachment sites (*attB*) were located in the SAP1 vector but not in the chromosome.

The transfer of the SAP1 vector containing the edited gene cluster for rapamycin was conducted as described in ref. [23]. The objective transconjugants were confirmed by the antibiotic-resistance phenotype and the size of the linear plasmid by contour-clamped homogeneous electric field (CHEF)[43] electrophoresis using SAP1::pKU503rap as the control linear plasmid.

**Cloning of *rapH* and expression in SUKA33, 34.** pKU565*tsr*-2cistron[23] was digested with *Hin*dIII and *Nhe*I, and then a small fragment was cloned into the same site of pKU592AT-*aac(3)*IV to give the expression vector pKU592AT-*aac(3) IV*-2cistron.

The *rapH* gene was amplified from pKU503rap as a template with PCR using a pair of primers (rapH_Xba_1F: GGCTCTAGATATGCCTGCCGTGGAGTGCTA TGAACTGGACGCCCG and rapH_Hind_1R: AAGAAGCTTCCTAGACGAGTT CGGCGGCAGGTGCTGGCGGCC). The amplified DNA fragment was digested with *Hin*dIII/*Xba*I and cloned into the same site of pKU592AT-*aac(3)IV*-2cistron to afford pKU592AT-2cis-*rapH*. The resultant plasmid was introduced into *S. avermitilis* SUKA33 and 34 by intergeneric conjugation and integration. The integration of the plasmid was confirmed based on its apramycin resistance and by PCR analysis. The resistance marker (*aac(3)IV*) flanking the *loxP* sequences was removed by the expression of a gene encoding Cre recombinase. The *cre*-expression plasmid, pKU471::*pheS*[A229G/T278A], was introduced into the exoconjugants by conjugation. The exonjugants were grown on yeast extract-malt extract medium[44] containing 2% D-xylose to induce the expression of *cre*. After sporulation, the 4-chlorophenylalanine-resistant and thiostrepton-sensitive and apramycin-sensitive progenies were selected, and the desired clones were defined as SUKA33::*rapH* and SUKA34::*rapH*, respectively.

**Cloning of *rapA*, *rapB* and *rapC*.** To prepare the PCR template, pKU503rap was digested with *Fsp*AI (ThermoFisher Scientific, Waltham, MA) to yield fragments containing *rapA* (27.0 kbp), *rapB* (28.2 kbp) and *rapC* (23.7 kbp), and each fragment was purified by CHEF electrophoresis. The largest fragment of *Nru*I-digested pKU518 was ligated with the purified *rapA*, *rapB* and *rapC* fragments to give pKU518rapA, pKU518rapB and pKU518rapC, respectively.

**Preparation of donor DNA fragments.** The donor DNA fragments were prepared by overlap PCR using a certain number of short fragments as template DNA. Each fragment was amplified by PCR with pKU518rapA, pKU518rapB or pKU518rapC as the appropriate template DNA using the primer pairs listed in Supplementary Tables 1 and 2. The resultant fragment DNAs were spliced by overlap PCR using the pairs of primers corresponding to the far ends.

**In vitro module editing of the rap gene cluster on BAC.** All sgRNAs were prepared using the EnGen sgRNA Synthesis Kit, *S. pyogenes* (New England Biolabs Japan Inc., Tokyo, Japan), according to the manufacturer's instructions. The target sequences are listed in Supplementary Table 2. For the in vitro Cas9 reaction, pKU503rap, a pair of sgRNAs (see Supplementary Table 1) and Cas9 (New England Biolabs Japan Inc., Tokyo, Japan) were mixed according to the manufacturer's instructions, and the reaction mixture was incubated at 37 °C for 120 min. The excision of the desired DNA fragment was confirmed by agarose gel electrophoresis (Fig. 1c). The purified BAC segment and the donor DNA fragment were subjected to Gibson assembly. After 50 min of incubation at 50 °C, the reaction mixture was treated with phenol/chloroform/isoamyl alcohol and then subjected to 2-propanol precipitation. The ligated DNA was transformed into *E. coli* NEB10 by electroporation. The transformants were selected on LA containing 100 µg ml$^{-1}$ ticarcillin. The desired plasmids were further selected by colony PCR using a pair of primers listed in Supplementary Table 1 and confirmed by restriction digestion.

**UPLC/UV/MS analysis of the metabolites from the transformant.** Five millilitres of culture broth were extracted with 5 ml of *n*-butanol (*n*-BuOH), 1.5 ml of the *n*-BuOH extract was dried in vacuo, and the residue was dissolved in 400 µl of dimethyl sulfoxide. Analytical UPLC and HR-ESI-MS (positive mode) were performed using a Waters ACQUITY UPLC System (Waters, Taunton, MA) in conjunction with a BEH ODS column (2.1 i.d. × 100 mm, Waters), a Waters ACQUITY UPLC photodiode array eλ detector (Waters), and a XevoG2 ToF system (Waters). Mobile phase A was water +0.1% formic acid, and mobile phase B was acetonitrile +0.1% formic acid. The elution program was 5 − 100% B over 5 min and 90% B for 1 min at a flow rate of 0.8 ml min$^{-1}$. Data acquisitions and analyses were performed by using MassLynX V4.1 software (Waters). The MS/MS analysis was performed with the Orbitrap Fusion Lumos Tribrid system (Thermo Fisher Scientific). Data acquisitions and analyses were performed by using Xcalibur V4.0.27.42 software (Thermo).

**Treatment of rapamycin with trifluoroacetic acid and water.** To a solution of rapamycin (1.0 mg) in CH$_2$Cl$_2$ (0.5 ml) at −40 °C was added trifluoroacetic acid (20 µl) followed by a drop of water. After 10 min and the reaction mixture was partitioned between ethyl acetate and saturated aqueous sodium bicarbonate. Layers were separated and organic layer successively washed with brine and dried over anhydrous sodium sulphate. The resulting crude material was subjected to UPLC-MS and MS/MS analyses.

**Determination of the configuration of 17.** To confirm the configuration of 16-hydroxyl, we synthesised a diastereomeric mixture of 16-*O*-demethyl rapamycin from **1** by the treatment with trifluoroacetic acid and water (see above). Also, using module editing technique, we obtained the deletion mutant of *rapM*, which

encodes the 16-*O*-specific methyltransferase[31,45], to prepare (16*S*)-16-*O*-demethyl-rapamycin (**20**) (Supplementary Data 85–91; indicating the same planar structure to **17**). UPLC analysis revealed that **17** was identical to the former peak of the diastereomeric mixture of 16-*O*-demethyl rapamycin (Supplementary Fig. 9) of which (16*R*)-epimer shows earlier retention time than the (16*S*)-epimer[30]. As expected, **20** was identical to the latter peak, the (16*S*)-epimer (Supplementary Fig. 9). Thus, we concluded that **17** is (16*R*)-16-*O*-demethyl rapamycin.

## Isolation of compounds

*Isolation of 2 from SUKA34::rapH/pKU503rapΔM9AT::M6ATm.* Four litres of fermentation broth of SUKA34::*rapH*/pKU503rapΔM9AT::M6ATm was centrifuged to obtain a mycelial cake, which was extracted twice with 500 ml of acetone. The acetone was removed in vacuo, and the residual aqueous layer was extracted twice with ethyl acetate (EtOAc). The resultant EtOAc layer was concentrated in vacuo to afford 1.9 g of crude extract. The crude extract was subjected to medium-pressure liquid chromatography (MPLC) on silica gel (SNAP Ultra 25 g, Biotage, Uppsala, Sweden) eluted with a gradient system of *n*-hexane–EtOAc (0–25% EtOAc) followed by a stepwise solvent system of chloroform (CHCl$_3$)–methanol (MeOH) (0, 1, 3, 5, 10, 50 and 90% MeOH). The 3% MeOH fraction (242 mg) was collected and subjected to silica gel MPLC (SNAP Ultra 25 g) with isocratic elution with 3% MeOH in CHCl$_3$. The fractions were monitored by UPLC analysis, and fractions containing **2** were collected (48.1 mg). The sample was further purified by preparative reversed-phase HPLC using a CAPCELL PAK MG-II C18 column (5.0 µm, 20 i.d. × 150 mm; Shiseido, Tokyo, Japan). **2** was detected using a 2996 photodiode array detector (Waters) and a 3100 mass detector (Waters) following elution with 80% aqueous acetonitrile, and 4.4 mg of **2** was obtained.

*Isolation of 6 from SUKA34::rapH/pKU503rapΔM8AT::M6ATm.* Six litres of fermentation broth of SUKA34::*rapH*/pKU503rapΔM8AT::M6ATm was centrifuged to obtain the mycelial cake, which was extracted twice with 500 ml of acetone. The acetone was removed in vacuo, and the residual aqueous layer was extracted twice with EtOAc. The resultant EtOAc layer was concentrated in vacuo to afford 3.3 g of crude extract. The crude extract was subjected to MPLC on silica gel (SNAP Ultra 25 g) eluted with a gradient system of *n*-hexane–EtOAc (0–25% EtOAc) followed by a stepwise gradient of CHCl$_3$–MeOH (0, 1, 3, 5, 10, 50 and 90% MeOH). The 3% MeOH fraction (441 mg) was collected and subjected to silica gel MPLC (SNAP Ultra 25 g) with isocratic elution with 3% MeOH in CHCl$_3$. The fractions were monitored by UPLC analysis, and fractions containing **6** were collected (209 mg). The sample was further purified by preparative reversed-phase HPLC using a CAPCELL PAK MG-II C18 column (5.0 µm, 20 i.d. × 150 mm; Shiseido) with 70% aqueous acetonitrile, yielding 1.6 mg of **6**.

*Isolation of 7 from SUKA33::rapH/pKU503rapΔM6.* Three litres of fermentation broth of SUKA33::*rapH*/pKU503rapΔM6 were centrifuged to obtain a mycelial cake, which was extracted twice with 500 ml of acetone. The acetone was removed in vacuo, and the residual aqueous layer was extracted twice with EtOAc. The resultant EtOAc layer was concentrated in vacuo to afford 1.77 g of crude extract. The crude extract was subjected to silica gel MPLC (SNAP Ultra 25 g, Biotage) eluted with a gradient system of *n*-hexane–EtOAc (0–25% EtOAc) followed by a stepwise gradient of CHCl$_3$–MeOH (0, 1, 5, 10, 50 and 90% MeOH). The 5% MeOH fraction (269 mg) was collected and subjected to silica gel MPLC (SNAP Ultra 25 g) with isocratic elution with 5% MeOH in CHCl$_3$. The fractions were monitored by UPLC analysis, and fractions containing **7** were collected (184 mg). The sample was further purified by preparative reversed-phase HPLC using a CAPCELL PAK MG-II C18 column (5.0 µm, 20 i.d. × 150 mm; Shiseido) with 75% aqueous acetonitrile as the eluent, and 4.5 mg of **7** was obtained.

*Isolation of 9 from SUKA33::rapH/pKU503rapΔM10.* Three litres of fermentation broth of SUKA33::*rapH*/pKU503rapΔM10 was centrifuged to obtain a mycelial cake, which was extracted twice with 500 ml of acetone. The acetone was removed in vacuo, and the residual aqueous layer was extracted twice with EtOAc. The resultant EtOAc layer was concentrated in vacuo to afford 1.67 g of crude extract. The crude extract was subjected to silica gel MPLC (SNAP Ultra 25 g, Biotage) eluted with a gradient system of *n*-hexane–EtOAc (0–25% EtOAc) followed by a stepwise gradient of CHCl$_3$–MeOH (0, 1, 5, 10, 50, 90% MeOH). The 5% MeOH fraction (345 mg) was collected and subjected to silica gel MPLC (SNAP Ultra 25 g) with isocratic elution with 5% MeOH in CHCl$_3$. The fractions were monitored by UPLC analysis, and the fractions containing **9** were collected (174 mg). The sample was further purified by Sephadex LH-20 column chromatography (1:1 CHCl$_3$/ CH$_3$OH) to obtain a fraction containing **9** (72.6 mg), which was then subjected to preparative reversed-phase HPLC using a CAPCELL PAK MG-II C$_{18}$ column (5.0 µm, 20 i.d. × 150 mm; Shiseido) with 65% aqueous acetonitrile a the eluent, and 12 mg of **9** was obtained.

*Isolation of 10 from SUKA33::rapH/pKU503rapΔM3.* Six litres of fermentation broth of SUKA33::*rapH*/pKU503rapΔM3, was centrifuged to obtain a mycelial cake, which was extracted twice with 500 ml of acetone. The acetone was removed in vacuo, and the residual aqueous layer was extracted twice with EtOAc. The

resultant EtOAc layer was concentrated in vacuo to afford 2.67 g of crude extract. The crude extract was subjected to silica gel MPLC (SNAP Ultra 25 g, Biotage) eluted with a gradient of $n$-hexane–EtOAc (0–25% EtOAc) followed by a stepwise gradient of $CHCl_3$–MeOH (0, 1, 3, 5, 10, 20% MeOH). The 3% MeOH fraction (382 mg) was collected and subjected to silica gel MPLC (SNAP Ultra 25 g) with isocratic elution with 3% MeOH in $CHCl_3$. The fractions were monitored by UPLC analysis, and the fractions containing **10** were collected (193 mg). The sample was subjected to Sephadex LH-20 column chromatography (1:1 $CHCl_3/CH_3OH$) and afforded a fraction containing **10** (45.9 mg). This sample was further purified by preparative reversed-phase HPLC using a CAPCELL PAK MG-II $C_{18}$ column (5.0 μm, 20 i.d. × 150 mm; Shiseido), and elution with 80% aqueous acetonitrile supplemented with 0.1% formic acid yielded 4.7 mg of **10**.

*Isolation of 11 from SUKA33::rapH/pKU503rapΔM2ACP-ΔM4KR.* Four litres of fermentation broth of SUKA33::*rapH*/pKU503rapΔM3, was centrifuged to obtain a mycelial cake, which was extracted twice with 500 ml of acetone. The acetone was removed in vacuo, and the residual aqueous layer was extracted twice with EtOAc. The resultant EtOAc layer was concentrated in vacuo to afford 2.79 g of crude extract. The crude extract was subjected to silica gel MPLC (SNAP Ultra 25 g, Biotage) eluted with a gradient of $n$-hexane–EtOAc (0–25% EtOAc) followed by a stepwise gradient of $CHCl_3$–MeOH (0, 1, 3, 5, 10, 20% MeOH). The 3% MeOH fraction (341.2 mg) was collected and subjected to silica gel MPLC (SNAP Ultra 25 g) with isocratic elution with 3% MeOH in $CHCl_3$. The fractions were monitored by UPLC analysis, and the fractions containing **11** were collected (70.1 mg). The sample was subjected to Sephadex LH-20 column chromatography (1:1 $CHCl_3/CH_3OH$) and afforded a fraction containing **11** (14.8 mg). This sample was further purified by preparative reversed-phase HPLC using a CAPCELL PAK MG-II $C_{18}$ column (5.0 μm, 20 i.d. × 150 mm; Shiseido), and elution with 80% aqueous acetonitrile supplemented with 0.1% formic acid yielded 4.7 mg of **11**.

*Isolation of 12 from SUKA33::rapH/pKU503rapΔM2AT-ΔM8AT.* Four litres of fermentation broth of SUKA34::*rapH*/ΔM2AT-ΔM8AT was centrifuged to obtain the mycelial cake, which was extracted twice with 500 ml of acetone. The acetone was removed in vacuo, and the residual aqueous layer was extracted twice with EtOAc. The resultant EtOAc layer was concentrated in vacuo to afford 1.29 g of crude extract. The crude extract was subjected to MPLC on silica gel (SNAP Ultra 25 g) eluted with a gradient system of n-hexane–EtOAc (0–25% EtOAc) followed by a stepwise gradient of $CHCl_3$–MeOH (0, 1, 3, 5, 10% MeOH). The 5% MeOH fraction (41.1 mg) was collected and further purified by Sephadex LH-20 column chromatography (1:1 $CHCl_3/CH_3OH$) to obtain a fraction containing **12** (28.8 mg). The sample was further purified by preparative reversed-phase HPLC using a CAPCELL PAK MG-II $C_{18}$ column (5.0 μm, 20 i.d. × 150 mm; Shiseido), and elution with 60% aqueous acetonitrile supplemented with 0.1% formic acid yielded 8.3 mg of **12**.

*Isolation of 13 from SUKA33::rapH/pKU503rap_M2-M12-M3.* Twelve point five litres of fermentation broth of SUKA33::*rapH*/pKU503rap_M2-M12-M3, was centrifuged to obtain a mycelial cake, which was extracted twice with 500 ml of acetone. The acetone was removed in vacuo, and the residual aqueous layer was extracted twice with EtOAc. The resultant EtOAc layer was concentrated in vacuo to afford 7.82 g of crude extract. The crude extract was subjected to silica gel MPLC (SNAP Ultra 25 g, Biotage) eluted with a gradient of $n$-hexane–EtOAc (0–25% EtOAc) followed by a stepwise gradient of $CHCl_3$–MeOH (0, 1, 3, 5, 10% MeOH). The 5% MeOH fraction (2.35 g) was collected and subjected to silica gel MPLC (SNAP Ultra 25 g) with stepwise gradient of $CHCl_3$–MeOH (0, 2, 3, 5% MeOH). The 3% MeOH fraction (638 mg) was subjected to Sephadex LH-20 column chromatography (1:1 $CHCl_3/CH_3OH$) and afforded a fraction containing **13** (66.5 mg). This sample was further purified by preparative reversed-phase HPLC using a CAPCELL PAK MG-II $C_{18}$ column (5.0 μm, 20 i.d. × 150 mm; Shiseido), and elution with 80% aqueous acetonitrile supplemented with 0.1% formic acid yielded 5.0 mg of **13**.

*Isolation of 14 from SUKA54/pKU503rap_M7ER[0].* pKU592AT-2cis-*rapH* was introduced into SUKA54/pKU503rap_M7ER[0] by intergeneric conjugation and integration to afford SUKA54::*rapH*/pKU503rap_M7ER[0]. Six litres of fermentation broth of SUKA54::*rapH*/pKU503rap_M7ER[0] was centrifuged to obtain a mycelial cake, which was extracted twice with 500 ml of acetone. The acetone was removed in vacuo, and the residual aqueous layer was extracted twice with EtOAc. The resultant EtOAc layer was concentrated in vacuo to afford 2.92 g of crude extract. The crude extract was subjected to silica gel MPLC (SNAP Ultra 25 g, Biotage) eluted with a gradient system of $n$-hexane–EtOAc (0–25% EtOAc) followed by a stepwise gradient of $CHCl_3$–MeOH (0, 1, 5, 10, 20, 50 and 90% MeOH). The 5% MeOH fraction (561 mg) was collected and subjected to ODS MPLC (Purif-Pack ODS-100) by using a $H_2O$-MeOH stepwise gradient (20, 40, 60, 80, and 100% MeOH). Fractions containing **14** were collected (227 mg). Then, 104.7 mg of this sample was further purified by preparative reversed-phase HPLC using a CAPCELL PAK MG-II $C_{18}$ column (5.0 μm, 20 i.d. × 150 mm; Shiseido), and elution with 75% aqueous acetonitrile yielded 7 mg of **14**.

*Isolation of 15 from SUKA34::rapH/pKU503rapΔM9DH-KR::M7DH-ER-KR.* Two point eight litres of fermentation broth of SUKA34::*rapH*/pKU503rapΔM9DH-

KR::M7DH-ER-KR, was centrifuged to obtain a mycelial cake, which was extracted twice with 500 ml of acetone. The acetone was removed in vacuo, and the residual aqueous layer was extracted twice with EtOAc. The resultant EtOAc layer was concentrated in vacuo to afford 1.44 g of crude extract. The crude extract was subjected to silica gel MPLC (Purif-pack SI-60, 27 g; Shoko Scientific) eluted with a gradient of $n$-hexane–EtOAc (0–25% EtOAc) followed by a stepwise gradient of $CHCl_3$–MeOH (0, 1, 3, 5, 10% MeOH). The 3% MeOH fraction (194 mg) was collected and subjected to silica gel MPLC (SNAP Ultra 25 g) with isocratic elution with 3% MeOH in $CHCl_3$. The fractions were monitored by UPLC analysis, and the fractions containing **15** were collected (110 mg). The sample was subjected to Sephadex LH-20 column chromatography (1:1 $CHCl_3/CH_3OH$) and afforded a fraction containing **15** (18.3 mg). This sample was further purified by preparative reversed-phase HPLC using a CAPCELL PAK MG-II $C_{18}$ column (5.0 μm, 20 i.d. × 150 mm; Shiseido), and elution with 85% aqueous acetonitrile yielded 3.2 mg of **15**.

*Isolation of 16 from SUKA34::rapH/pKU503rapΔM3DH-ER-KR::M7DH-ER-KR.* Seven litres of fermentation broth of SUKA34::*rapH*/pKU503rapΔM3DH-ER-KR:: M7DH-ER-KR, was centrifuged to obtain a mycelial cake, which was extracted twice with 500 ml of acetone. The acetone was removed in vacuo, and the residual aqueous layer was extracted twice with EtOAc. The resultant EtOAc layer was concentrated in vacuo to afford 3.83 g of crude extract. The crude extract was subjected to silica gel MPLC (SNAP Ultra 25 g, Biotage) eluted with a gradient system of $n$-hexane–EtOAc (0–25% EtOAc) followed by a stepwise gradient of $CHCl_3$–MeOH (0, 1, 3, 5, 10% MeOH). The 3% MeOH fraction (144 mg) was collected and subjected to Sephadex LH-20 column chromatography (1:1 $CHCl_3/CH_3OH$) and afforded a fraction containing **16** (55.7 mg). This sample was further purified by preparative reversed-phase HPLC using a CAPCELL PAK MG-II $C_{18}$ column (5.0 μm, 20 i.d. × 150 mm; Shiseido), and elution with 80% aqueous acetonitrile supplemented with 0.1% formic acid yielded 10.6 mg of **16**.

*Isolation of 17 from SUKA34::rapH/pKU503rapΔM11DHL-KR::bfmM2KR.* Six litres of fermentation broth of SUKA34::*rapH*/pKU503rapΔM11DHL-KR::bfmM2KR was centrifuged to obtain a mycelial cake, which was extracted twice with 500 ml of acetone. The acetone was removed in vacuo, and the residual aqueous layer was extracted twice with EtOAc. The resultant EtOAc layer was concentrated in vacuo to afford 2.75 g of crude extract. The crude extract was subjected to silica gel MPLC (SNAP Ultra 25 g, Biotage) eluted with a gradient of $n$-hexane–EtOAc (0–25% EtOAc) followed by a stepwise gradient of $CHCl_3$–MeOH (0, 1, 3, 5, 10, 50 and 90% MeOH). The 5% MeOH fraction (228 mg) was collected and then subjected to Sephadex LH-20 column chromatography (1:1 $CHCl_3/CH_3OH$) and afforded a fraction containing **17** (33.2 mg). This sample was further purified by preparative reversed-phase HPLC using a CAPCELL PAK MG-II $C_{18}$ column (5.0 μm, 20 i.d. × 150 mm; Shiseido), and elution with 60% aqueous acetonitrile yielded 2.2 mg of **17**.

*Isolation of (16S)-16-O-demethyl rapamycin (20) from SUKA34::rapH/pKU503rapΔrapM.* One litre of fermentation broth of SUKA34::*rapH*/pKU503rapΔrapM was centrifuged to obtain a mycelial cake, which was extracted twice with 500 ml of acetone. The acetone was removed in vacuo, and the residual aqueous layer was extracted with EtOAc twice. The resultant EtOAc layer was concentrated in vacuo to afford 0.74 g of crude solid. The crude extract was subjected to silica gel MPLC (SNAP Ultra 25 g, Biotage) eluted with a gradient of $n$-hexane–EtOAc (0–25% EtOAc) followed by a stepwise gradient of $CHCl_3$–MeOH (0, 1, 3, 5, 10, 50 and 90% MeOH). The 5% MeOH fraction (59.8 mg) was collected and then subjected to Sephadex LH-20 column chromatography (1:1 $CHCl_3/CH_3OH$) and afforded a fraction containing **20** (6.4 mg). This sample was further purified by preparative reversed-phase HPLC using a CAPCELL PAK MG-II $C_{18}$ column (5.0 μm, 20 i.d. × 150 mm; Shiseido), and elution with 78% aqueous MeOH yielded 2.9 mg of **20**.

**Binding activities of rapamycin derivatives to FKBP and mTOR.** HeLa cells were maintained in DMEM (Fujifilm Wako, Osaka, Japan) supplemented with 10% FBS (Thermo Fisher Scientific, MA, USA) and 1% of Penicillin-Streptomycin (Thermo Fisher Scientific) in a humidified incubator with 5% $CO_2$ at 37 °C. Harvested cells were plated at $1 \times 10^6$ cells/well in 6-well plate and incubated for 6 h. Then each 1.25 μg of phAG-mTOR and pAsh-FKBP12 (Medical and Biological Laboratories Co., Ltd., Aichi, Japan) was co-transfected with Lipofectamine 2000 reagent (Thermo Fisher Scientific) overnight. Next day, transfected cells were harvested and seeded in 384-well optical-bottom microplate (PerkinElmer, MA, USA) at a density of 3000 cells/well. After the 24 h incubation at 37 °C, cells were treated with different concentrations of compounds for 1 h. Cells were then fixed with 4% paraformaldehyde and stained with 1 μg/ml of Hoechst33342 (Thermo Fisher Scientific). Cell images of 9 visual fields/well were acquired with ×20 objective lens in Opera Phenix High-Content Screening System (PerkinElmer) and analysed by Harmony4.9 software (PerkinElmer). Because protein-protein interactions are quantified by measuring the aggregation of fluorescent elements in foci, fluorescent intensity in foci was normalised by total fluorescent intensity in each cell. $EC_{50}$ values were calculated using TIBCO Spotfire 10.3 software (PerkinElmer).

**Cytotoxicity assay.** The cytotoxic activities of isolated rapamycin derivatives against T lymphoma Jurkat cells, human ovarian adenocarcinoma SKOV-3 cells

and malignant pleural mesothelioma MESO-1 cells were examined. Jurkat cells were cultured in RPMI1640 medium supplemented with 10% foetal bovine serum, penicillin (50 U/ml), streptomycin (50 μg/ml), and Glutamax. SKOV-3 cells were cultured in DMEM medium supplemented with 10% fetal bovine serum, penicillin (50 U/ml), and streptomycin (50 μg/ml). MESO-1 cells were cultured in RPMI1640 medium supplemented with 10% fetal bovine serum, penicillin (50 U/ml), and streptomycin (50 μg/ml). All cell lines were seeded in a 384-well plate at a density of 1000 cells/well in 20 μl of media and incubated at 37 °C in a humidified incubator with 5% $CO_2$. After 4 h, 2-fold dilution samples resolved in DMSO were added to the cell culture at the concentration of 0.5% (0.1 μl) and incubated for 72 h. Cell viabilities were measured using a CellTiter-Glo luminescent cell viability assay and EnVision multilabel plate reader.

**Reporting summary**. Further information on research design is available in the Nature Research Reporting Summary linked to this article.

## Data availability

Sequence data that support the findings of this study have been deposited in DDBJ with the accession code LC566301. Source data are provided with this paper. The other data that support the findings of this study are included in this paper. Raw data for spectroscopic analyses are available from the corresponding author upon reasonable request.

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

## Acknowledgements

This work was supported by Japan Agency for Medical Research and Development (AMED) under Grant Number JP19ae0101045 for K.S.

## Author contributions

Conceptualisation, K.K., T.H. and K.S.; data curation, K.K. and T.H.; funding acquisition, K.S.; investigation, K.K., T.H., J.H., I.K., N.K., R.U. and T.N.; methodology, K.K., T.H., M. K., H.I. and K.S.; project administration, K.S.; resources, H.I. and K.S.; Supervision, H.I. and K.S.; validation, K.K., T.H., H.S. and K.S.; visualisation, K.K. and T.H.; Writing–original draft, K.K., T.H. and K.S.; Writing–review and editing, K.K., T.H., H.S. and K.S.

## Competing interests

The authors declare no competing interests.
