## [Peer Review File · Nature Communications]

Reviewers' Comments:

Reviewer #1:

Remarks to the Author:

This manuscript describes an in vitro Cas9 based method to edit the genes of type I PKSs. After cutting by Cas9, Gibson assembly is used to provide the desired engineered PKS gene, and the plasmid is transformed into a heterologous host and the target compound isolated. This is a very nice approach to mutating PKS genes that leverages the ability of Cas9 to site-selectively cut even very large BACs. This will surely be a nice addition to the PKS engineering toolbox. However, enthusiasm is reduced due to some caveats that are not made clear and the otherwise lack of novelty as currently presented. For example, the approach inherently relies on the abilities of the heterologous host to provide the necessary building blocks, and on the potential permissiveness of tailoring enzymes. In addition, It is not particularly clear which compounds accessed here are actually novel. It would be helpful to cite literature that has reported other rapamycin analogues for comparison. Further, the production of compounds 17-19 requires further explanation. How is the stereochemistry of the hydroxyl in 17 determined? Where does the methylation event in 18 derive from? In general, types of exchanges similar to those described here have been reported many times before. Thus, the emphasis here is not on the rational design of modular PKSs as the title suggests, but on simply leveraging Cas9 to make the desired sequence changes in vitro. Very little has been learned with respect to the catalytic abilities of the PKS modules, especially given that the absolute or relative yields of each product has not been explicitly described. In addition, nothing has been learned with respect to optimal boundaries for domain/module swapping. It seems that most of the boundaries and mutagenesis was designed on the basis of prior literature precedent. While quite useful, this method will likely be insufficient to generate new analogs at-will unless problems with protein:protein interactions and substrate specificity/channeling have also been solved. Accordingly, I don't recommend publication in its current form but would consider a heavily revised version of the manuscript.

Other issues that need to be addressed:

p2, Abstract: "Despite intense studies, the accurate targeting of the desired regions in the type-I modular polyketide synthase (PKS) gene has not been accomplished." This is exaggerated or inaccurate. There are hundreds of examples in the literature and patent literature that demonstrate the successful manipulation of type I PKSs.

p3, lines 1-2: "middle-weight molecular compounds have attracted increasing attention as new compounds to fill the gap between antibodies and low-molecular-weight compounds. This sentence needs to be edited to clarify what gap is being addressed, i.e. gap in molecular weight.

p3, line 3: "PPI" should be written in full.

p3, line 8: For clarity, please specify whether the difficulties cited are those of chemical synthesis and/or biosynthesis. The next sentence ("Therefore, various congeners will be produced from known useful natural compounds by genetic manipulation) doesn't follow naturally.

p3: "Relatively few experiments on gene modification in modular PKS genes have been reported." As stated, this is untrue.

p4: The introduction would benefit from including greater discussion on the mutagenesis and gene assembly methods available for type I PKSs and how this fits in with production of polyketides when the gene editing is done in vitro. This would better support the novelty of the CRISPR-Cas9 approach reported.

p4: "SUKA" needs writing in full or supporting with a citation. Please also further describe why these were chosen.

p5: "If we can accurately edit the modular PKS of rapamycin, which shows very high homology, this technique will be applicable for most natural products." This sentence needs clarifying. Is the Rap system selected here because the Rap modules show high homology to each other? Or do the Rap modules show high homology to other PKSs? In either case, some quantitative analysis would be helpful to support this argument.

p5: There are several areas that need clarifying in the "AT domain-targeted module editing" section. For example, "Our first aim was to exchange the extender unit for a particular module of rapamycin PKS by swapping the AT domains from other modules". It's not possible to replace and extender unit (acyl-CoA) with a module. In another example, "We first targeted module 9 to methylate the triene moiety of rapamycin" doesn't make sense. The AT domain does not methylate the triene. Rather, a malonyl-CoA is replaced with a methylmalonyl-CoA at this position in the polyketide. After this, please make it clearer that each AT domain targeted is being replaced with AT domains from the same Rap PKS, but different modules. Similarly, it is not clear what "targeted" means until SI Fig. 2 is viewed. It may be useful to the reader to define that Mod6AT and Mod8AT are being inserted at these positions.

p15: "Therefore, we attempted to edit multiple modules". Suggest adding "simultaneously" for clarification.

p19. Table 1: Please include error bars and standard deviations. Please also unify the use of significant figures.

p20: "we believe that the substrate specificities of the PKS modules are much broader than expected". I don't see any evidence in this manuscript that supports this statement, particularly without yields or quantification of the each product. Please clarify or remove. There are plenty of other published examples that do support PKS promiscuity.

SI Figure 6: Why are the HPLC peaks for compound 8 and 17 not visible?

Reviewer #2:

Remarks to the Author:

This manuscript by Kudo et al. describes the combined application of in vitro CRISPR-Cas9 and Gibson assembly technology to edit type 1 modular PKSs. Authors demonstrated that the method developed can be applied for the engineering AT domain, module deletion/insertion, reductive domains, stereochemistry of beta-hydroxyl group, and double modification (ER and KR) using rapamycin PKS as a model system. Although in vitro CIRPR/Cas9 system for refactoring of biosynthetic gene clusters in *Streptomyces* has been reported previously (mBio 6(6):e01714-15), this manuscript may be the first report to use CIRPR/Cas9 technology to edit the large modular PKSs.

Although the method used in this study seems promising to edit PKS efficiently, authors should present the complete structural data (e.g. 1D and 2D NMR) of all compounds generated in this study to prove the accuracy and efficiency of the method as claimed in the manuscript. Some compounds might be produced in a small amount. However, authors should show that this method at least produces a reasonable amounts of the designed polyketide for structure determination to prove their claims that this in vitro editing method can produce almost all the designed compounds and modular PKSs have broader substrate specificities than expected. The structures of many of the new compounds generated in this study were predicted only by MS fragmentation pattern, which may not be acceptable in a high-profile journal such as Nature Communications.

1. Abstract is too general to provide the actual outcome of this study.

2. Page 3, line 4: What is PPI? Please give the full name.
3. Page 3, last two lines: "Relatively few experiments on gene modification in modular PKS genes have been reported." This sentence is not true. There are a large number of papers on the engineering modular PKS. The main problem in PKS engineering is that it is difficult to predict the results due to the complex structure of giant PKS protein and there is no rule to design the recombinant PKS.
4. Page 4, the second line from the bottom: Please give the reference of "SUKA" strain here where it appears first.
5. Page 5, line 10: Please describe the quantitative production level of SUKA33::rapH or SUKA34::rapH.
6. Page 5, last line: "We first targeted module 9 to methylate the triene moiety of rapamycin which is reported to play a significant role in the biological activity." Please give reference of this fact.
7. Page 7: Please describe the quantitative production levels of compounds 2, 3, 4, 6, and 6 in the main text, although they were described in the SI. Please do the same for other compounds. Importantly, NMR data of compounds 2 and 6 were provided in the SI. It is critical to validate the chemical structure of all analogs obtained in this study for proving the efficiency of the method developed in this study. Although Supplementary Figure 4 shows the MS fragmentation patterns of compounds 3, 4, and 5, all structure should be confirmed by NMR.
8. Page 9: The NMR data of compound 8 should be provided.
9. Page 10: The NMR data of compounds 10, 11, 12, and 13 should be provided.
10. Page 13: The NMR data of compounds 15 and 16 should be provided.
11. Page 16: The NMR data of compounds 18 and 19 should be provided.
12. Page 17: "Biological activities". Unfortunately, all bioactivities of new compounds were reduced. It is strongly suggest to test their immunosuppressive activity to see if some of new analogs showing applicable antifungal activity (e.g. compound 17) can have potential to develop as antifungal agents with reduced immunosuppressive activity.
13. Authors should describes how many independent experiments were carried out to present all the chromatograms in SI.

Reviewer #3:

Remarks to the Author:

The manuscript describes a strategy for using Cas9 and guide RNAs to site-specifically modify a polyketide synthase gene cluster in vitro. Domains/modules were replaced by desiging guide RNAs that flank these regions, using Cas9 to impart double strand breaks on either side, and a replacement sequence was added using PCR/gibson assembly.

These modified PKS clusters, housed within BAC heterologous expression vectors, were then transformed into a heterologous host and the resulting compounds were analyzed. The changes induced at the genetic level resulted in expected modifications to the chemical structures of the products. The authors thoroughly characterized the resulting products using mass spectrometry, multi-dimensional NMR, and UV-VIs spectroscopy. This reviewer appreciated the explanations of which key signals in the NMR spectra support their conclusions that were in provided in addition to full spectral assignments in the SI.

In general, the authors provide convincing data to support their conclusions. One exception to this is the experiments targeting inversion of stereochemistry (pg.15). If the full NMR characterization of this derivative is still forthcoming, perhaps it would be best to report that in a subsequent manuscript.

Genetic modification of the rapamycin gene cluster to produce derivative compounds is not a novel achievement, but the exact methodology employed here, using cas9 in vitro on clusters in BACs, is novel as far as this reviewer knows. The work presented will be of interest in the field.

One suggested improvement would be to provide a detailed/quantitative comparison of the mutation incidence rate of this methodology compared with previously reported techniques that are described in the introduction of this manuscript. Another suggested improvement would be to provide information about the titer/yield of the various rapamycin derivatives. The authors provide convincing evidence that they induced the expected changes in chemical structures of products, but if the genetic modifications result in a significant decrease in yield of derivatives compared to "wild-type" rapamycin, this information is highly relevant.

We thank you for generously supporting our manuscript revision.

We have revised our manuscript as follows.

Authors' possible answers (A) to the comments (C) raised by Reviewer #1.

Reviewer #1 (Remarks to the Author):

This manuscript describes an in vitro Cas9 based method to edit the genes of type I PKSs. After cutting by Cas9, Gibson assembly is used to provide the desired engineered PKS gene, and the plasmid is transformed into a heterologous host and the target compound isolated. This is a very nice approach to mutating PKS genes that leverages the ability of Cas9 to site-selectively cut even very large BACs. This will surely be a nice addition to the PKS engineering toolbox.

We appreciate the evaluation of our methodology. As this reviewer pointed out, how to edit very large and extremely homologous sets of genes, such as type I PKS genes, is a major issue that requires a solution.

(C-1) However, enthusiasm is reduced due to some caveats that are not made clear and the otherwise lack of novelty as currently presented. For example, the approach inherently relies on the abilities of the heterologous host to provide the necessary building blocks, and on the potential permissively of tailoring enzymes.

(A-1) We usually utilize three engineered hosts (SUKA, *S. lividans* derived strains, and *S. albus* derived strains) to generate microbial secondary metabolites of Actinomycetes origin. Using these hosts, we accomplished heterologous expression with a success rate of more than 60% without additional gene modifications, such as the insertion of promoters or transcriptional regulators (these modifications dramatically improved the heterologous expression success rate, as was the case for telomestatin (*Sci. Rep.* **7**, 3382 (2017)) and quinolidomicin (*Org. Lett.* **20**, 7996-7999 (2018); ref.14). Of course, each host appears to preferably produce certain compounds. Among these hosts, we mainly use SUKA hosts, which are derived from the industrial producer of avermectin. We have proven that SUKA hosts can produce a wide range of compounds, as reported in ref. 17. In addition to those compounds, we also achieved the heterologous expression of more than 50 compounds, including oligomycin, bottromycin, toyocamycin, and telomestatin. In contrast, *S. lividans*-derived hosts can produce quinolidomicin, the largest macrolide of terrestrial origin, and borrelidin. Therefore, we strongly believe that our technique in these hosts is applicable for a wide range of compounds, and we would select different hosts based on their production yields. In this study, we utilized the SUKA strain (in this strain, the yield of rapamycin was 8.6 mg/L, while that of the *S. lividans*-derived strain was 1.9 mg/L.).

We have added a sentence describing why SUKA strains were chosen as hosts on page 5,

lines 15-17.

(C-2) In addition, It is not particularly clear which compounds accessed here are actually novel. It would be helpful to cite literature that has reported other rapamycin analogues for comparison.

(A-2) We have indicated whether the compounds mentioned were newly produced. We cited a *Nature Communications* paper (ref. 10) as an index of already reported analogues. Additionally, we have cited a *Med. Chem. Comm.* paper (2018) that reviews the synthetic analogues of rapamycin (ref. 18; page 5, lines 10-11).

(C-3) Further, the production of compounds 17-19 requires further explanation. How is the stereochemistry of the hydroxyl in 17 determined?

(A-3) We have added a supplementary figure to further explain (through further analyses) how we determined the stereochemistry of the hydroxyl in compound **17** (Supplementary Fig. 9, explained on page 16, lines 14-16). It has been reported that its 16(S)-epimer is less polar than its 16(R)-epimer (*J. Org. Chem.* **59**, 6512-6513 (1994); ref. 29). We prepared a mixture of 16(S)-O-demethyl rapamycin and 16(R)-O-demethyl rapamycin based on the literature. The MS/MS spectra of 16(S)-O-demethyl rapamycin and 16(R)-O-demethyl rapamycin were identical, which indicated that their planar structures are the same (i.e., the 16(S)- and 16(R)-epimers were synthesized). Then, we performed HPLC analysis to compare the retention time of compound **17** against that of chemically synthesized 16-O-demethyl rapamycin. As expected, the retention time of compound **17** was identical to that of the primary peak of the chemically synthesized standard, which corresponds to the 16(R)-epimer. Together with these results, we again concluded that compound **17** is (16R)-16-O-demethyl-rapamycin.

(C-4) Where does the methylation event in 18 derive from?

(A-4) The origin of the methyl group of compound **18** is discussed in the revised manuscript on page 17, lines 8-12. The relaxed substrate specificity of RapM (*Chem. Sci.* **6**, 2885-2892 (2015); ref. 30) may cause the unpredictable methylation event in engineered polyketide substrates.

(C-5) In general, types of exchanges similar to those described here have been reported many times before. Thus, the emphasis here is not on the rational design of modular PKSs as the title suggests, but on simply leveraging Cas9 to make the desired sequence changes in vitro. Very little has been learned with respect to the catalytic abilities

of the PKS modules, especially given that the absolute or relative yields of each product has not been explicitly described. In addition, nothing has been learned with respect to optimal boundaries for domain/module swapping. It seems that most of the boundaries and mutagenesis was designed on the basis of prior literature precedent. While quite useful, this method will likely be insufficient to generate new analogs at-will unless problems with protein:protein interactions and substrate specificity/channeling have also been solved. Accordingly, I don't recommend publication in its current form but would consider a heavily revised version of the manuscript.

(A-5) We are sorry, but we believe that the importance of our claims was not made fully clear to Reviewer #1 and we strongly refute this reviewer's comment. Who could manipulate such a large type-I PKS biosynthetic gene cluster to strictly target the desired regions? As we described in this manuscript, former studies were based on restriction enzymes or homologous recombination techniques. There are a large number of restriction enzyme sites in giant biosynthetic genes and many homologous regions in giant type-I PKS biosynthetic gene clusters. Therefore, the previous study on rapamycin module swapping (*Nat. Commun.*, **8**, 1206 (2017); ref. 10) did not accomplish the authors' initial purpose, although attractive results were provided. Thus, we thought to apply our BAC vector and heterologous expression technology and succeeded in overcoming the bottleneck in the manipulation of large type-I PKS biosynthetic gene clusters (we produced the exact compound (compound **16**) that the authors of the *Nat. Commun.* paper aimed to prepare.). We have already carried out module editing against several type-I PKS compounds and NRPS compounds, which we will publish in the near future.

The aim of this study was to establish a methodology for the flexible design of type I PKSs but not to suggest a novel theory for protein engineering. Many papers reporting improvements in PKS design at the protein level have been published, but the technology to produce engineered PKS genes has not been updated. In this context, we have established a credible methodology to edit rapamycin PKS genes and showed it to be a good replacement for recombination-based editing. Since even the protein designs based on the prior literature can produce novel analogues for the assessment of biological activity, future theoretical improvements (designs) will surely accelerate the derivatization of potential natural products. Technological advances for the sound construction and expression of designed genes are indispensable to these future improvements. Thus, our report of gene editing combined with heterologous expression will be used for the drug discovery process in the future. Based on these points, we have changed the title of the paper to "Rational Editing of Modular Polyketide Synthase Genes to Produce Desired

Natural Product Derivatives".

Other issues that need to be addressed:

(C-6) p2, Abstract: "Despite intense studies, the accurate targeting of the desired regions in the type-I modular polyketide synthase (PKS) gene has not been accomplished." This is exaggerated or inaccurate. There are hundreds of examples in the literature and patent literature that demonstrate the successful manipulation of type I PKSs.

(A-6) We have changed this sentence to clarify this issue.

(C-7) p3, lines 1-2: "middle-weight molecular compounds have attracted increasing attention as new compounds to fill the gap between antibodies and low-molecular-weight compounds. This sentence needs to be edited to clarify what gap is being addressed, i.e. gap in molecular weight.

(A-7) We have added the phrases "in molecular weight" and "macromolecules, such as" on page 3, line 3, to clarify the gap that has been addressed.

(C-8) p3, line 3: "PPI" should be written in full.

(A-8) We have defined this abbreviation (page 3, line 5).

(C-9) p3, line 8: For clarity, please specify whether the difficulties cited are those of chemical synthesis and/or biosynthesis. The next sentence ("Therefore, various congeners will be produced from known useful natural compounds by genetic manipulation) doesn't follow naturally.

(A-9) We have added the words "by chemical synthesis" on page 3, line 10.

(C-10) p3: "Relatively few experiments on gene modification in modular PKS genes have been reported." As stated, this is untrue.

(A-10) We have changed this sentence to "Although many works on type I PKS modification have been performed, only a few pioneering works have succeeded in modifying modular PKS genes of larger type I PKS compounds to generate new analogues of targeted polyketides" (page 3, last line to page 4, lines 1-2).

(C-11) p4: The introduction would benefit from including greater discussion on the mutagenesis and gene assembly methods available for type I PKSs and how this fits in with production of polyketides when the gene editing is done in vitro. This would

better support the novelty of the CRISPR-Cas9 approach reported.

(A-11) We thank the reviewer for this comment. We have added a sentence on page 4, lines 9-13.

(C-12) p4: “SUKA” needs writing in full or supporting with a citation. Please also further describe why these were chosen.

(A-12) The word is a proper noun defined in a paper published in 2010 (*Proc. Natl. Acad. Sci. U. S. A.* **107**, 2646-51 (2010)).

(C-13) p5: “If we can accurately edit the modular PKS of rapamycin, which shows very high homology, this technique will be applicable for most natural products.” This sentence needs clarifying. Is the Rap system selected here because the Rap modules show high homology to each other? Or do the Rap modules show high homology to other PKSs? In either case, some quantitative analysis would be helpful to support this argument.

(A-13) Rap modules show high homology to each other. We have clarified this point in the manuscript and provided quantitative data regarding the average homology between KS domains on page 5, lines 12-13. Because it is difficult to determine the sequence homology between entire modules due to differences in the domain organization of each PKS system, we compared only the KS domains. Every module in each type-I PKS biosynthetic gene cluster is similar each other, but that of rapamycin is much more homologous.

(C-14) p5: There are several areas that need clarifying in the “AT domain-targeted module editing” section. For example, “Our first aim was to exchange the extender unit for a particular module of rapamycin PKS by swapping the AT domains from other modules”. It’s not possible to replace and extender unit (acyl-CoA) with a module. In another example, “We first targeted module 9 to methylate the triene moiety of rapamycin” doesn’t make sense. The AT domain does not methylate the triene. Rather, a malonyl-CoA is replaced with a methylmalonyl-CoA at this position in the polyketide. After this, please make it clearer that each AT domain targeted is being replaced with AT domains from the same Rap PKS, but different modules. Similarly, it is not clear what “targeted” means until SI Fig. 2 is viewed. It may be useful to the reader to define that Mod6AT and Mod8AT are being inserted at these positions.

(A-14) We thank you for reading carefully. We have corrected these sentences on page 6, lines 9-14.

(C-15) p15: “Therefore, we attempted to edit multiple modules” . Suggest adding “simultaneously” for clarification.

(A-15) We have added this word based on your comment.

(C-16) p19. Table 1: Please include error bars and standard deviations. Please also unify the use of significant figures.

(A-16) In order to evaluate the biological activities of our derivatives including **10**, **11**, **13**, **15** and **16** (additionally isolated compounds for this revision), we performed the cytotoxicity assay again. At the same time, we also observed the PPI-inductive activities between mTOR and FKBP of these compounds by employing Fluoppi (Fluorescent based technology detecting Protein-Protein Interactions) analysis. As the results, we found that the binding activities of **15** simultaneously to FKBP12 and mTOR was as potent as rapamycin while the cytostatic effect was ~100-fold weakened. Based on this observation, the “Biological activities” section has been entirely reconstructed. In the revised version, Table 1 was replaced with EC₅₀ values calculated from Fluoppi data, and the results of viability test was moved to Supplementary Figure 10 which includes error bars indicating standard deviations for each plot. Unfortunately, since the yield of **6** was so low that we did not have enough amount of **6** to perform the assays, we have omitted **6** from this section.

(C-17) p20: “we believe that the substrate specificities of the PKS modules are much broader than expected” . I don’ t see any evidence in this manuscript that supports this statement, particularly without yields or quantification of the each product. Please clarify or remove. There are plenty of other published examples that do support PKS promiscuity.

(A-17) We have changed the sentence to be more objective; it now discusses our results within the context of previous examples of PKS promiscuity (page 21, lines 12-14). Additionally, we have added Supplementary Table 3 which shows the yield of each product.

(C-18) SI Figure 6: Why are the HPLC peaks for compound 8 and 17 not visible?

(A-18) These peaks are not visible because the vertical axis of each chromatogram was adjusted according to the highest peak in each figure. Compound **8** was the derivative in our series of experiments produced in the lowest yield and could be detected only by MS spec. The HPLC peak for compound **17** was much broader than those for the other derivatives. In the revised Supplementary Figure 6, the HPLC chromatogram for compound **17** is magnified so readers can more easily see the broad peak.

Authors' possible answers (A) to the comments (C) raised by Reviewer #2.

Reviewer #2 (Remarks to the Author):

This manuscript by Kudo et al. describes the combined application of in vitro CRISPR-Cas9 and Gibson assembly technology to edit type 1 modular PKSs. Authors demonstrated that the method developed can be applied for the engineering AT domain, module deletion/insertion, reductive domains, stereochemistry of beta-hydroxyl group, and double modification (ER and KR) using rapamycin PKS as a model system. Although in vitro CIRPR/Cas9 system for refactoring of biosynthetic gene clusters in *Streptomyces* has been reported previously (mBio 6(6):e01714-15), this manuscript may be the first report to use CIRPR/Cas9 technology to edit the large modular PKSs.

We thank Reviewer #2 for the positive evaluation of our technology. Since the engineering of PKS genes does not allow even a single base error, precise and theoretically robust methodology is required.

(C-1) Although the method used in this study seems promising to edit PKS efficiently, authors should present the complete structural data (e.g. 1D and 2D NMR) of all compounds generated in this study to prove the accuracy and efficiency of the method as claimed in the manuscript. Some compounds might be produced in a small amount. However, authors should show that this method at least produces a reasonable amount of the designed polyketide for structure determination to prove their claims that this in vitro editing method can produce almost all the designed compounds and modular PKSs have broader substrate specificities than expected. The structures of many of the new compounds generated in this study were predicted only by MS fragmentation pattern, which may not be acceptable in a high-profile journal such as *Nature Communications*.

(A-1) We have added NMR data for compounds **10, 11, 12, 13, 15** and **16**, whose yields were high enough allow their analysis by NMR. In addition, we confirmed the quantitative analysis of all compounds (Supplementary Table 3). According to the results, compounds **3, 4, 8** and **18** were produced, but their yields were much lower than those of the other compounds. Isolation of compound **5** was tried, however, pure compound **5** with enough quality for NMR was not obtained due to instability of **5**. We could detect degradation products from compound **5** during isolation process. (Fig. A(a), shown below). Those degradation products showed similar UV spectrum pattern to **5** (Figs. A(b) and (c)) and 14 smaller *m/z* than that of **5** (Figs. A(d) and (e)). Multiple degradation products were detected (Fig. B) and those peaks were not detectable from crude extract before isolation. We measured a NMR spectrum of the collected sample (ca 0.8 mg), however, the sample

still included impurities for structural elucidation (Fig. C).

Figure A. Degradation of **5** under the isolation condition.

Figure B. Mass chromatogram at 908.51 of the collected fraction.

Figure C. ^1H NMR spectrum of the collected fraction.

Since the production yield of compound **19** was small (0.23 mg/L), we performed large scale culture. Unfortunately, compound **19** was so unstable that we could not isolate it even after several rounds of culture.

We understand that this reviewer's comment follows the manuscript submission policy; therefore, we would like to introduce these compounds and their yields. If necessary, we may be able to increase production of the compound of interest by further optimization of domain/module boundaries, gene expression, culture conditions and so on. Because information on accessibility to new skeletons is important, we think that it is worth showing what can be produced.

(C-2) 1. Abstract is too general to provide the actual outcome of this study.

(A-2) Changes were made to more specifically describe the features of our study and the outcomes of this study.

(C-3) 2. Page 3, line 4: What is PPI? Please give the full name.

(A-3) We have defined this abbreviation (protein-protein interaction) on page 3, line 5.

(C-4) 3. Page 3, last two lines: "Relatively few experiments on gene modification in modular PKS genes have been reported." This sentence is not true. There are a large number of papers on the engineering modular PKS. The main problem in PKS engineering is that it is difficult to predict the results due to the complex structure of giant PKS protein and there is no rule to design the recombinant PKS.

(A-5) We have replaced the sentence with a more appropriate one (page 3, last line to page 4, lines 1-2). Additionally, we have added a sentence on page 4, lines 9-13 that further discussed the problems in PKS engineering and clarified what we improved. Because there are "obstacles due to the high sequence similarities of PKS modules/domains", it was very difficult to precisely manipulate a giant PKS gene cluster as large as rapamycin PKS until our technology was used instead of the methods previously employed.

(C-5) 4. Page 4, the second line from the bottom: Please give the reference of "SUKA" strain here where it appears first.

(A-5) We have added a reference (ref. 17).

(C-6) 5. Page 5, line 10: Please describe the quantitative production level of SUKA33::rapH or SUKA34::rapH.

(A-6) We have added information on the quantitative production of SUKA34::rapH (page 6,

lines1-2).

(C-7) 6. Page 5, last line: “We first targeted module 9 to methylate the triene moiety of rapamycin which is reported to play a significant role in the biological activity.” Please give reference of this fact.

(A-7) We have added a reference to the crystal structure of the ternary complex of human FKBP12 (FK506-binding protein), rapamycin, and the FRB (FKBP12-rapamycin-binding, a part of mTOR) domain (ref. 21).

(C-8: comments from 7 to 12) 7. Page 7: Please describe the quantitative production levels of compounds 2, 3, 4, 6, and 6 in the main text, although they were described in the SI. Please do the same for other compounds. Importantly, NMR data of compounds 2 and 6 were provided in the SI. It is critical to validate the chemical structure of all analogs obtained in this study for proving the efficiency of the method developed in this study. Although Supplementary Figure 4 shows the MS fragmentation patterns of compounds 3, 4, and 5, all structure should be confirmed by NMR.

8. Page 9: The NMR data of compound 8 should be provided.

9. Page 10: The NMR data of compounds 10, 11, 12, and 13 should be provided.

10. Page 13: The NMR data of compounds 15 and 16 should be provided.

11. Page 16: The NMR data of compounds 18 and 19 should be provided.

(A-8) Response for items 7-11: Data on the quantitative production of all compounds have been added (Supplementary Table 3). In addition, the NMR data for compounds 10, 11, 12, 13, 15 and 16 are provided.

(C-9) 12. Page 17: “Biological activities” . Unfortunately, all bioactivities of new compounds were reduced. It is strongly suggest to test their immunosuppressive activity to see if some of new analogs showing applicable antifungal activity (e.g. compound 17) can have potential to develop as antifungal agents with reduced immunosuppressive activity.

(A-9) A recently reported analogue of rapamycin, DL001 (*Nat. commun.*, **10**, 3194 (2019)), which was prepared by organic synthesis, was reported to show selective effects against mTOR1, reducing immunosuppressive activities. In a similar fashion, rapamycin derivatives may show other biological activities beyond their immunosuppressive and antitumor activities. As the reviewer suggested, we are also interested in other biological activities of these derivatives, since derivatives of rapamycin as well as FK506 did not show high immunosuppressive activity (At the Fujisawa Pharmaceutical Company, the

immunosuppressive activities of many FK506 derivatives were tested, but natural compound FK506 showed the highest immunosuppressive activity. These results strongly suggested the difficulties in developing derivatives to overcome clinically utilized compounds.). Therefore, we would like to compare the immunosuppressive and the cytotoxic activities of newly produced derivatives, but unfortunately, we do not have a system in which to observe these immunosuppressive effects.

Instead, we performed the Fluoppi (Fluorescent based technology detecting Protein-Protein Interactions) experiment which detects the formation of FKBP-rapamycin-mTOR ternary complex (supplied by Medical & Biological Laboratories Co., Ltd., Aichi, Japan). We examined all isolated derivatives except for **6** which was the least produced derivative. According to the result, the binding activities of **15** simultaneously to FKBP12 and mTOR was turned out to be as potent as rapamycin while the cytostatic effect was ~100-fold weakened (we have been aware that rapamycin shows the cytostatic effect rather than the cytotoxicity). Therefore, we concluded that the showing IC₅₀ values is not appropriate for both rapamycin and most of its derivatives. Based on this observation, the "Biological activities" section has been entirely reconstructed. In the revised version, Table 1 was replaced with EC₅₀ values calculated from Fluoppi data, and the results of viability test was moved to Supplementary Figure 10.

Since these changes have made the interpretation clearer and we could recognize the interesting activity of **15**, we appreciate this reviewer's suggestions.

(C-10) 13. Authors should describes how many independent experiments were carried out to present all the chromatograms in SI.

(A-10) We have added information about the number of independent experiments performed.

Authors' possible answers (A) to the comments (C) raised by Reviewer #3.

Reviewer #3 (Remarks to the Author):

The manuscript describes a strategy for using Cas9 and guide RNAs to site-specifically modify a polyketide synthase gene cluster in vitro. Domains/modules were replaced by designing guide RNAs that flank these regions, using Cas9 to impart double strand breaks on either side, and a replacement sequence was added using PCR/gibson assembly. These modified PKS clusters, housed within BAC heterologous expression vectors, were then transformed into a heterologous host and the resulting compounds were analyzed. The changes induced at the genetic level resulted in expected modifications to the chemical structures of the products. The authors thoroughly characterized the resulting products using mass spectrometry, multi-dimensional NMR, and UV-VIS spectroscopy. This reviewer appreciated the explanations of which key signals in the NMR spectra support their conclusions that were in provided in addition to full spectral assignments in the SI.

We thank this reviewer for providing these positive comments. The fact that the designed derivatized skeleton of a large polyketide could be produced is one of the most important results we would like to emphasize.

(C-1) In general, the authors provide convincing data to support their conclusions. One exception to this is the experiments targeting inversion of stereochemistry (pg.15). If the full NMR characterization of this derivative is still forthcoming, perhaps it would be best to report that in a subsequent manuscript.

(A-1) We have added Supplementary Fig. 9 to further explain (through further analyses) how we determined the stereochemistry of compound **17**. Using *J. Org. Chem.* (1994), we compared the retention times of the 16(*S*)-epimer (natural configuration) and 16(*R*)-epimer (compound **17**) and confirmed that compound **17** is the 16(*R*)-epimer.

(C-2) Genetic modification of the rapamycin gene cluster to produce derivative compounds is not a novel achievement, but the exact methodology employed here, using cas9 in vitro on clusters in BACs, is novel as far as this reviewer knows. The work presented will be of interest in the field.

(A-2) We thank this reviewer again for evaluating the novelty of our methodology. We are pleased to introduce a precise way to manipulate the PKS cluster.

(C-3) One suggested improvement would be to provide a detailed/quantitative comparison of the mutation incidence rate of this methodology compared with previously reported

techniques that are described in the introduction of this manuscript.

(A-3) We apologize, but it is difficult to directly compare two strategies because all clones obtained by the recombination-based technique were not intended. The PKS construct initially targeted in the previous paper was successfully obtained in our experiment and shown to be functional for the first time, leading to compound **16**.

(C-4) Another suggested improvement would be to provide information about the titer/yield of the various rapamycin derivatives. The authors provide convincing evidence that they induced the expected changes in chemical structures of products, but if the genetic modifications result in a significant decrease in yield of derivatives compared to "wild-type" rapamycin, this information is highly relevant.

(A-4) The yield of each compound has been added (Supplementary Table 3).

Reviewers' Comments:

Reviewer #1:

Remarks to the Author:

The authors have addressed the majority of suggestions and criticisms of this reviewer. Revision of the introduction, title, and several other sections of the manuscript have shifted the focus towards how the edits were introduced, as opposed to focusing on PKS engineering itself. This is a welcome shift. With this in mind, I would still like to see a small change to the title - it should mention Cas9 or CRISPR to emphasize more readily the actual technological advance that is being described. I believe this would increase readership significantly. In addition, I suggest mention of earlier works that similarly use Cas9 to achieve gene editing in vitro. This would better place the work in context and properly acknowledge previous advances that are relevant here.

<https://doi.org/10.1128/mBio.01714-15>

<https://doi.org/10.1038/ncomms9101>

It might also be prudent to cite earlier works that describe employing Cas9 directly in producing strains so reader can decide for themselves whether carrying out editing in a heterologous host vs native strain is preferred.

Reviewer #2:

Remarks to the Author:

Authors addressed some concerns raised by reviewers. However, there are a few issues that should be addressed more carefully.

Complete NMR data of all compounds were requested, however only six of them were included in the revised version, which might be acceptable due to the low production and instability observed. Nevertheless, the determination of the stereochemistry of hydroxyl group at C16 position of compound 17, which was also reviewer #1's concern, should have been addressed more carefully. In Supplementary Figure 9, authors chemically synthesized racemic mixture of (16S) and (16R)-16-O-demethyl rapamycin and compared their retention time with that of compound 17. Although Supplementary reference 6 showed the preparation of 16S) and (16R)-16-O-demethyl rapamycin, there are three methoxy groups in rapamycin and the chemical demethylation may not be so specific, the methoxy groups at C16 and C27 positions may be chemically equivalent. Authors should confirm the structure of 16-O-demethyl rapamycin, which was used as a standard for compound 17, by NMR.

In addition, if authors can explain the possible cause of instability of compound 5, it will be better. For this reviewer, it is not easy to know the reason of instability of compound 5.

New Supplementary Table 3 showing the yield of rapamycin derivatives obtained in this study clearly says that about half of the derivatives were produced in trace amounts. This might be due to the inadequate fusion sites between PKS domains. Although this kind of low production from the engineered PKS is not unusual, in order to prove the utility of the method developed in this study, I strongly suggest to improve the production levels of some derivatives by construction of several new engineered PKSs. This may not be successful to increase the yield, however authors can at least show the utility of the new method to engineer the large PKS. This should not be too difficult if the new method is efficient and promising as authors claimed.

Finally, although authors modified the sentence on the previous studies on the engineering modular PKSs in introduction (line 48-50), this can still be misleading. It is suggested to review previous relevant studies more broadly but concisely and describe the advantage the method shown in this study.

Reviewer #3:

Remarks to the Author:

The revisions presented by the authors have addressed this reviewer's critiques.

We thank all reviewers again for generously supporting our manuscript revision.

We have revised our manuscript as follows.

Authors' possible answers (A) to the comments (C) raised by Reviewer #1.

(C-1') The authors have addressed the majority of suggestions and criticisms of this reviewer. Revision of the introduction, title, and several other sections of the manuscript have shifted the focus towards how the edits were introduced, as opposed to focusing on PKS engineering itself. This is a welcome shift. With this in mind, I would still like to see a small change to the title - it should mention Cas9 or CRISPR to emphasize more readily the actual technological advance that is being described. I believe this would increase readership significantly. In addition, I suggest mention of earlier works that similarly use Cas9 to achieve gene editing in vitro. This would better place the work in context and properly acknowledge previous advances that are relevant here.

<https://doi.org/10.1128/mBio.01714-15>

<https://doi.org/10.1038/ncomms9101>

It might also be prudent to cite earlier works that describe employing Cas9 directly in producing strains so reader can decide for themselves whether carrying out editing in a heterologous host vs native strain is preferred.

(A-1') We appreciate this reviewer for giving many fruitful suggestions that made the manuscript much clearer than that of the original state.

We have added the words "in vitro Cas9-assisted" on the top of the manuscript's title.

We have cited the references given by this reviewer and added some sentences on page 21, line 7-11. Also, we have cited two references reviewing the systems for in vivo genome editing of *Streptomyces*. As mentioned on page 21, line 17 to page 22, line 1-2, it is still challenging to set an efficient protocol for each producing strain in interest. In general, low DNA transformation efficiency of producing strains than the model strains frequently brings problems. Also, as a genomic DNA sizes roughly 8-10 Mb in *Streptomyces*, the chance of off-target recognition by Cas9 complex is more frequent compared to using a BAC clone of which size spans 100-200 kb.

Authors' possible answers (A) to the comments (C) raised by Reviewer #2.

(C-1') Authors addressed some concerns raised by reviewers. However, there are a few issues that should be addressed more carefully.

Complete NMR data of all compounds were requested, however only six of them were included in the revised version, which might be acceptable due to the low production

and instability observed. Nevertheless, the determination of the stereochemistry of hydroxyl group at C16 position of compound 17, which was also reviewer #1's concern, should have been addressed more carefully. In Supplementary Figure 9, authors chemically synthesized racemic mixture of (16S) and (16R)-16-O-demethyl rapamycin and compared their retention time with that of compound 17. Although Supplementary reference 6 showed the preparation of (16S) and (16R)-16-O-demethyl rapamycin, there are three methoxy groups in rapamycin and the chemical demethylation may not be so specific, the methoxy groups at C16 and C27 positions may be chemically equivalent. Authors should confirm the structure of 16-O-demethyl rapamycin, which was used as a standard for compound 17, by NMR.

(A-1') We appreciate this reviewer for understanding of low productivity problems.

As the Supplementary reference 6 reported, the chemical demethylation at the C16 position is specific because the neighbouring conjugated triene promotes a series of chemical reactions presented in the paper. Thus, the methoxy groups at C16 is more reactive than those of the C27 and C39 positions.

Whereas, we also think that additional data that reviewer recommended is also precious to further proof. Just in case, to support the position of the demethylation, we prepared (16S)-16-O-demethyl rapamycin from the culture broth of SUKA34/pKU503rap Δ rapM, which is the deletion mutant of the gene encoding the 16-O-specific methyltransferase (Gregory, M. A. et al., *Org. Biomol. Chem.* (2006) and Law, B. J. C. et al., *Chem. Sci.* (2015)), and confirmed its structure by NMR (Supplementary data 85-91). Then we performed HPLC analysis again, confirming the purified (16S)-16-O-demethyl rapamycin was identical to the (16S)-epimer prepared from chemical conversion (see the revised version of Supplementary Fig. 9). Thus, the demethylation position was determined as C16. Accordingly, we concluded that the other epimer obtained from chemical conversion is the diastereomer of 16-hydroxyl group (i.e. (16R)-16-O-demethyl rapamycin).

With these, we have changed the manuscript on page 16, line14-15, and added a subsection to Supplementary Information text on page 7.

(C-2') In addition, if authors can explain the possible cause of instability of compound 5, it will be better. For this reviewer, it is not easy to know the reason of instability of compound 5.

(A-2') One possible explanation is the stability of the trienyl carbocation derived from the elimination of the methoxyl group at the C16 position. If the structure of compound 5 is as we designed, the methyl group at the C23 position of rapamycin is substituted to a hydrogen atom. This C-H_b bond (see the scheme below) can interact with the trienyl moiety to form

hyperconjugation, while the C-H_a bond cannot. In comparison, the C-Me bond of rapamycin does not stabilise the trienyl carbocation by forming hyperconjugation. Thus, since the trienyl carbocation derived from compound 5 is more stable than that of rapamycin, the elimination of the methoxy group at the C16 position is promoted. We consider that a nucleophilic attack by a water molecule to the trienyl carbocation may result in the significant degradation product of which *m/z* value was 908.

(C-3') New Supplementary Table 3 showing the yield of rapamycin derivatives obtained in this study clearly says that about half of the derivatives were produced in trace amounts. This might be due to the inadequate fusion sites between PKS domains. Although this kind of low production from the engineered PKS is not unusual, in order to prove the utility of the method developed in this study, I strongly suggest to improve the production levels of some derivatives by construction of several new engineered PKSs. This may not be successful to increase the yield, however authors can at least show the utility of the new method to engineer the large PKS. This should not be too difficult if the new method is efficient and promising as authors claimed.

(A-3') We agree that the fusion site is very important for the productivities of compounds.

Indeed, we have already tested some varieties of fusion sites as described below to see the effect on the production level of one derivative not mentioned in the manuscript. As a result, we could observe just a slight difference between the fusion sites tested (please see the figure below and note that this is closed data). Of course, there is the possibility that the residue by residue investigation may result in the robust improvement of the production. However, it is a much bigger experiment or not the focus of this study. We think it is crucial to improve the production level as this reviewer pointed out, but, in our opinion, the intensive design of tailor-made PKS for drug development still requires some theoretical advances in PKS enzyme engineering.

*** This is CLOSED data***

Closed data figure redacted at authors' request

*** This is CLOSED data ***

(C-4') Finally, although authors modified the sentence on the previous studies on the engineering modular PKSs in introduction (line 48-50), this can still be misleading. It is suggested to review previous relevant studies more broadly but concisely and describe the advantage the method shown in this study.

(A-4') We have changed the sentence into "With these contexts, many have tried to modify type I PKS to produce new analogues of targeted polyketides and proposed several genetic

strategies, which still have potential limitations on applying them to larger type I PKS compounds." We claim that it is essential not to depend on the recombination-based or restriction site (short recognition site nuclease) for manipulating large-sized PKS genes.

Authors' possible answers (A) to the comments (C) raised by Reviewer #3.

(C-1') The revisions presented by the authors have addressed this reviewer's critiques.

(A-1') We thank this reviewer again for giving positive evaluations and spending much time for us.

Reviewers' Comments:

Reviewer #2:

Remarks to the Author:

Authors addressed all concerns raised by reviewers, and this manuscript can be accepted for publication now. I appreciate authors' effort to make this study complete.